# Fractional is Better: Learnable Derivative Orders in Neural Operator Learning

**Fares B. Mehouachi** [1]  **Saif Eddin Jabari** [1 2]

## Abstract

Neural operators learn mappings between function spaces, enabling fast surrogate solutions to partial differential equations. Despite remarkable architectural diversity, these methods often share a common input representation: raw coordinate-value pairs $(x, u(x))$. We ask whether inputs aligned with PDE differential structure can improve learning. Through Picard iteration on mild solutions, we show that derivatives of the input appear explicitly in the solution operator, suggesting that providing derivative features should reduce the network's implicit differentiation burden. We prove this intuition: providing derivative features improves approximation rates from $O(W^{-(s-m)/d})$ to $O(W^{-s/d})$, where $W$ is network width, $s$ is input regularity, $m$ is the PDE order, and $d$ is spatial dimension. Our central finding, however, is a surprise: the optimal derivative order $\beta^*$ is strictly less than the PDE order $m$. This gap arises from a bias-variance tradeoff in spectral space that we characterize in closed form. Learning $\beta$ from data achieves automatic spectral regularization. We introduce $\partial$-NO (del-NO), for derivative-augmented neural operators, an architecture-agnostic augmentation that provides learnable fractional derivative features to any neural operator backbone. Across benchmark problems and architectures, $\partial$-NO consistently improves prediction accuracy, with learned orders that reflect a representation of known physics modulated by noise and finite-sample constraints.

## 1. Introduction

Neural operators have emerged as a powerful tool for scientific computing. By learning mappings between function spaces from data, they enable fast surrogate solutions to partial differential equations, with applications spanning climate modeling, materials science, and engineering design (Li et al., 2020; Lu et al., 2021; Kovachki et al., 2023). The field has witnessed rapid architectural innovation: spectral convolutions (Li et al., 2020), attention mechanisms (Cao, 2021; Hao et al., 2023), physics-aware designs (Wu et al., 2024), and latent representations (Wu et al., 2023). These advances have substantially improved accuracy and efficiency. Yet beneath this diversity lies a striking uniformity. Virtually all neural operators represent their inputs in the same way:

$$\text{1D: } (x, u(x)), \quad \text{2D: } (x, y, u(x, y)). \tag{1}$$

This representation ignores the differential structure that defines the underlying physics. Partial differential equations are built from derivatives: $\partial_x u$, $\partial_{xx} u$, and their combinations. A neural operator receiving only function values must learn to differentiate implicitly from finite, noisy samples. This places an unnecessary burden on the network.

**This paper asks a simple question:** Can we provide inputs that align with PDE differential structure? And if so, what is the best way to do it?

Our answer proceeds in three steps. First, we analyze the structure of PDE solution operators through Picard iteration on mild solutions. This reveals that derivatives of the input function appear explicitly in the solution operator, so providing them as features reduces the implicit differentiation burden on the network (§3). Second, we prove that providing these features improves approximation rates by a factor depending on the PDE order (§4). Third, and most unexpectedly, we uncover an operator-level bias-variance tradeoff: the optimal derivative order $\beta^*$ satisfies

$$\boxed{\beta^* < m} \tag{2}$$

for any finite, noisy dataset (§5). Using the exact PDE order $m$ introduces estimation variance that outweighs the approximation benefit. The gap $m - \beta^*$ shrinks as data grows, but remains positive for any finite, noisy dataset.

This finding has important practical implications. It means derivative features should not simply match known physics, but should balance physical fidelity against statistical efficiency. Since this optimal order depends on unknown quan-

[1]New York University Abu Dhabi, UAE. [2]New York University, NY, USA. Correspondence to: Fares B. Mehouachi <fares.mehouachi@nyu.edu>.

*Proceedings of the 43rd International Conference on Machine Learning*, Seoul, South Korea. PMLR 306, 2026. Copyright 2026 by the author(s).

tities like noise level and sample size, we propose learning it directly from data.

The resulting architecture, which we call $\partial$-NO (del-NO) for derivative-augmented neural operators (Figure 1; §6), augments any backbone with fractional derivative features whose orders are jointly optimized during training. The method is simple to implement, adds minimal overhead, and wraps existing architectures without modification.

Where does $\partial$-NO sit among existing uses of fractional calculus in deep learning? At least three established lines exist: *fractional PINNs* include fractional operators directly in the loss (Pang et al., 2019; Guo et al., 2022; Ren et al., 2023; Javadi et al., 2023); *fractional gradient methods* replace integer gradients with fractional ones during optimization (Khan et al., 2018; Shin et al., 2023; Elnady et al., 2025); and *fractional architectures* embed fractional dynamics into the network itself (Pu et al., 2017; Cui et al., 2025; Kang et al., 2024; Coelho et al., 2024). $\partial$-NO occupies a distinct fourth role: learnable fractional derivative *input features* applied as preprocessing, with no modification to equation, optimizer, or architecture.

## 2. Background

### 2.1. Neural Operators

Let $\mathcal{G} : \mathcal{U} \to \mathcal{V}$ denote the true solution operator between function spaces. A neural operator $\mathcal{G}_\theta$ is its finite-dimensional approximation parameterized by $\theta$, learned from paired examples $\{(u_i, v_i)\}_{i=1}^n$ with $v_i = \mathcal{G}[u_i]$ by minimizing the empirical risk:

$$\min_\theta \frac{1}{n} \sum_{i=1}^n \|\mathcal{G}_\theta[u_i] - v_i\|_{L^2}^2. \tag{3}$$

The target operator is typically a PDE solution map, taking initial or boundary conditions to the corresponding solution.

Different architectures approach this problem from different angles. The Fourier Neural Operator (Li et al., 2020) performs convolutions efficiently in frequency space. Deep-ONet (Lu et al., 2021) factorizes the operator into branch and trunk networks. More recent work explores attention mechanisms (Hao et al., 2023; Wu et al., 2024; Luo et al., 2025) and implicit representations (Serrano et al., 2023). What these methods share is their input representation: they receive the input function sampled at spatial locations, either as coordinate-value pairs or on a regular grid.

### 2.2. A Concrete Example: Burgers' Equation

To make the limitation concrete, consider the viscous Burgers' equation:

$$\partial_t u + u\,\partial_x u = \nu\,\partial_{xx} u. \tag{4}$$

The solution operator $\mathcal{G} : u(\cdot, 0) \mapsto u(\cdot, T)$ maps initial conditions to solutions at time $T$. This operator depends on the initial condition through two mechanisms: the diffusive term $\nu\partial_{xx}u$ smooths the solution according to second derivatives, while the convective term $u\partial_x u$ transports features at speeds determined by first derivatives. A standard neural operator must learn both differentiation operations from data.

Recent work has shown empirically that providing precomputed derivatives as additional inputs improves neural operator performance (Zhu et al., 2025). This raises a natural question: which derivatives should we provide? When the PDE is known, one might supply $\partial_x u$ and $\partial_{xx} u$ to match the equation structure. But what if the governing physics is unknown or only partially specified? And even when known, is matching the PDE order necessarily optimal?

These considerations motivate a different approach: rather than prescribing derivative orders, let the network learn them from data. To enable this, we need derivatives defined for non-integer orders.

### 2.3. Fractional Derivatives

To enable optimization over derivative orders, we need a continuous parameterization. Fractional calculus provides exactly this.

**Definition 2.1** (Fractional Derivative). For order $\beta \geq 0$ and function $u \in H^s(\Omega)$ with $s > \beta$, we define the *Weyl fractional derivative* through the Fourier transform:

$$D^\beta u := \mathcal{F}^{-1}\left[(i\xi)^\beta \hat{u}(\xi)\right], \tag{5}$$

where $\hat{u}(\xi) = \mathcal{F}[u](\xi)$ denotes the Fourier transform and $(i\xi)^\beta = |\xi|^\beta e^{i(\pi/2)\beta \, \text{sgn}(\xi)}$.

The Weyl derivative acts as a frequency filter weighting mode $\xi$ by $(i\xi)^\beta$, preserving phase information that distinguishes odd and even derivatives. When $\beta = 1$, we recover $\partial_x$; when $\beta = 2$, we get $\partial_{xx}$. Intermediate values interpolate smoothly. For non-periodic domains or to avoid Fourier artifacts (Gibbs ringing), we use the Grünwald-Letnikov finite-difference formulation (Appendix H).

Crucially, the definition is differentiable with respect to $\beta$:

$$\frac{\partial}{\partial \beta}(i\xi)^\beta = (i\xi)^\beta \left(\log|\xi| + i\frac{\pi}{2}\text{sgn}(\xi)\right). \tag{6}$$

This enables gradient-based learning of derivative orders.

## 3. Picard Iteration and Derivative Emergence

Why should derivative features help neural operators? The answer emerges from analyzing how PDE solutions depend on initial conditions through Picard iteration.

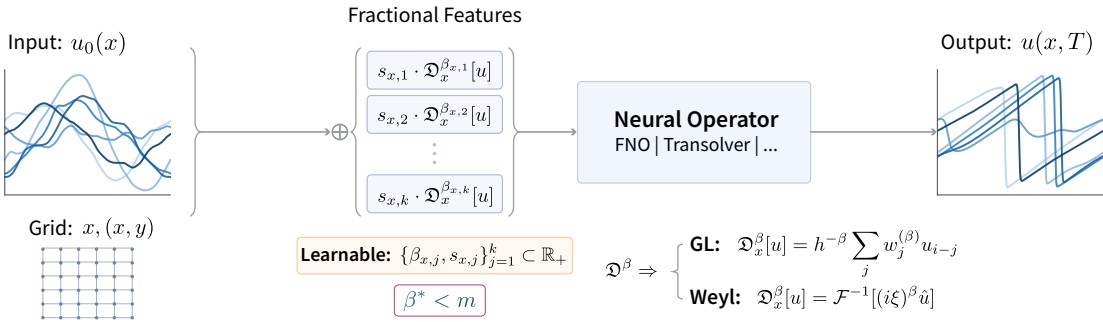

*Figure 1.* $\partial$-NO architecture. The input function $u$ is augmented with learnable fractional derivatives $\mathfrak{D}^{\beta_k} u$, where each order $\beta_k$ and scale $s_k$ is optimized jointly with backbone weights during training. This simple augmentation wraps any neural operator backbone without modification.

### 3.1. Mild Solutions and the Green's Function

Consider a semilinear PDE of the form:

$$\partial_t u = \mathcal{L}u + \mathcal{N}[u], \quad u(0) = u_0, \tag{7}$$

where $\mathcal{L}$ is a linear differential operator of order $m$ and $\mathcal{N}$ is a nonlinearity (polynomial or quasilinear). The mild solution satisfies the integral equation:

$$u(t) = G_t * u_0 + \int_0^t G_{t-s} * \mathcal{N}[u(s)] \, ds, \tag{8}$$

where $G_t$ is the Green's function (semigroup) generated by $\mathcal{L}$. For constant-coefficient operators, this takes the explicit form $G_t = e^{t\mathcal{L}}$.

### 3.2. The Picard Mechanism

Picard iteration constructs solutions by successive approximation. Starting from $u^{(0)}(t) = G_t * u_0$, each iterate refines the approximation:

$$u^{(k+1)}(t) = G_t * u_0 + \int_0^t G_{t-s} * \mathcal{N}[u^{(k)}(s)] \, ds. \tag{9}$$

The key observation is that the first iterate already reveals essential structure.

Expanding the semigroup for small times, $G_s * u_0 \approx u_0 + s\mathcal{L}u_0 + O(s^2)$, and substituting into the nonlinearity:

$$\mathcal{N}[G_s * u_0] \approx \mathcal{N}[u_0] + s \cdot D\mathcal{N}[u_0][\mathcal{L}u_0] + O(s^2), \tag{10}$$

where $D\mathcal{N}[u_0][v]$ is the Fréchet derivative of $\mathcal{N}$ at $u_0$ applied to direction $v$. Evaluating the integral yields:

**Theorem 3.1** (Derivative Emergence, Parabolic Case). *Let $\mathcal{G} : u_0 \mapsto u(T)$ be the solution operator for a semilinear parabolic PDE with polynomial nonlinearity $\mathcal{N}$ of degree $N$ and linear operator $\mathcal{L}$ of differential order $m$. For $u_0 \in B_s(R)$, there exists $T_*(R) = (L(R))^{-1}$, where $L(R)$ is the Lipschitz constant of $\mathcal{N}$ on $B_s(R)$, such that for all $T < T_*(R)$:*

$$\mathcal{G}[u_0] = G_T * u_0 + T \cdot \bar{G}_T * \mathcal{N}[u_0] + \mathcal{R}(u_0, T), \tag{11}$$

*where $\bar{G}_T = T^{-1} \int_0^T G_{T-s} \, ds$ is an averaged Green's function, and the remainder satisfies $\|\mathcal{R}(u_0, T)\|_{L^2} \leq C(R)T^2$ uniformly over $u_0 \in B_s(R)$. The nonlinearity $\mathcal{N}[u_0]$ involves derivatives of $u_0$ up to order $m$ and powers up to $u_0^N$.*

The full proof appears in Appendix B. The essential point is that derivatives enter through two mechanisms: the linear operator $\mathcal{L}$ directly differentiates $u_0$, while the nonlinearity $\mathcal{N}$ combines powers and derivatives according to its structure.

*Remark* 3.2 (Beyond Short Times). The derivative dependence persists for arbitrary times via iterated Picard and extends to stationary PDEs (e.g., Darcy flow) and quasilinear systems (e.g., Navier-Stokes) where derivatives appear directly. See Appendix B.2.

*Remark* 3.3 (Beyond Parabolic Evolution). A complementary spectral resolvent argument (Appendix B.3) extends the same derivative structure to general evolution equations $\partial_t^a u = \mathcal{L}u$ with temporal order $a > 0$, recovering parabolic ($a=1$), hyperbolic ($a=2$), and fractional-in-time (Mittag-Leffler) cases uniformly.

**Example 3.4** (Burgers Features). For Burgers' equation, $\mathcal{L} = \nu\partial_{xx}$ and $\mathcal{N}[u] = -u\partial_x u$. The solution operator depends on:

$$\{u_0, \partial_x u_0, \partial_{xx} u_0, u_0 \cdot \partial_x u_0\}. \tag{12}$$

The first three are derivatives; the fourth emerges from the convective nonlinearity. In this work, we focus on the *derivative terms* $\{u_0, \partial_x u_0, \partial_{xx} u_0\}$, leaving the nonlinear product $u_0 \cdot \partial_x u_0$ to the network. This differs from equation-aware approaches that encode full nonlinear structure (Zhu et al., 2025); our method requires only the PDE order $m$, not equation-specific terms.

## 4. Approximation Theory: Why Features Help

We now prove that derivative features improve approximation rates. This section establishes that providing derivative

features is beneficial; the *optimal* choice of derivative order is deferred to Section 5. The improvement quantifies how much easier the learning problem becomes when inputs align with PDE structure.

### 4.1. The Feature Map

**Definition 4.1** (Feature Maps). We define two related feature maps. For a single derivative order $\beta \geq 0$, the $\beta$-*feature map* $\Phi^\beta : H^s(\Omega) \to H^s \times H^{s-\beta}$ is:

$$\Phi^\beta[u] = (u, D^\beta u). \tag{13}$$

For the full set of integer derivatives up to order $m$, the *full feature map* $\Phi^{[m]} : H^s(\Omega) \to \mathcal{X}$ is:

$$\Phi^{[m]}[u] = (u, D^1 u, D^2 u, \ldots, D^m u), \tag{14}$$

where $\mathcal{X} = H^s \times H^{s-1} \times \cdots \times H^{s-m}$ with the product norm. When context is clear, we write $\Phi$ for either map.

The feature map bundles the input function with its derivatives. Since $\Phi$ includes $u$ as its first component, no information is lost.

**Assumption 4.2** (Regularity). Inputs lie in a Sobolev ball: $u \in B_s(R) := \{u \in H^s(\Omega) : \|u\|_{H^s} \leq R\}$ with $s > m + d/2$.

**Proposition 4.3** (Stability). *Under Assumption 4.2, the feature map $\Phi : B_s(R) \to \mathcal{X}$ is bi-Lipschitz:*

$$\|u_1 - u_2\|_{H^s} \leq \|\Phi[u_1] - \Phi[u_2]\|_{\mathcal{X}} \leq \sqrt{1+m} \cdot \|u_1 - u_2\|_{H^s}. \tag{15}$$

*The upper constant $\sqrt{1+m}$ is independent of the ball radius $R$.*

The lower bound is immediate since $\Phi$ preserves $u$. The upper bound follows from the continuity of differentiation in Sobolev spaces. See Appendix C for the full proof.

### 4.2. Main Approximation Result

**Theorem 4.4** (Approximation Rates). *Let $\mathcal{G} : B_s(R) \to L^2(\Omega)$ be the solution operator for a PDE of differential order $m$. Denote by $\mathcal{NO}_W$ the class of neural operators with $W$ parameters.*

*(i) Without features:*

$$\inf_{\mathcal{G}_\theta \in \mathcal{NO}_W} \sup_{u \in B_s(R)} \|\mathcal{G}[u] - \mathcal{G}_\theta(u)\|_{L^2} = O\left(W^{-\frac{s-m}{d}}\right). \tag{16}$$

*(ii) With full features $\Phi^{[m]}$:*

$$\inf_{\mathcal{G}_\theta \in \mathcal{NO}_W} \sup_{u \in B_s(R)} \left\|\mathcal{G}[u] - \mathcal{G}_\theta(\Phi^{[m]}[u])\right\|_{L^2} = O\left(W^{-\frac{s}{d}}\right). \tag{17}$$

*Proof.* Both bounds are achievability results (upper bounds on the minimax error). Full details appear in Appendix D.

**Part (i):** The operator $\mathcal{G} : B_s(R) \to L^2$ has modulus of continuity $\omega(\varepsilon) \leq C(R)\varepsilon^{(s-m)/s}$ when measured in $L^2$. This follows from Sobolev interpolation: for $u_1, u_2 \in B_s(R)$ with $\|u_1 - u_2\|_{L^2} \leq \varepsilon$, we have $\|u_1 - u_2\|_{H^m} \lesssim \varepsilon^{(s-m)/s} R^{m/s}$. The Hölder exponent $(s - m)/s$ arises because inputs lie in $H^s$ but the operator depends on $m$-th order derivatives: the "missing" regularity $m$ out of total $s$ determines the continuity loss. The $\epsilon$-covering number of $B_s(R)$ in $L^2$ satisfies $\log \mathcal{N}(\epsilon) \asymp (R/\epsilon)^{d/s}$ (Carl & Stephani, 1990). Applying the approximation theory of (Yarotsky, 2017) for Hölder-continuous functions on sets with this metric entropy yields the rate $W^{-(s-m)/d}$.

**Part (ii):** Define the lifted operator $\tilde{\mathcal{G}}[\Phi^{[m]}[u]] := \mathcal{G}[u]$. On bounded balls, $\tilde{\mathcal{G}}$ is Lipschitz: since the features $\Phi^{[m]}[u] = (u, D^1 u, \ldots, D^m u)$ provide the derivatives explicitly, the operator no longer needs to "invert" differentiation, an unbounded operation that was the source of the Hölder discontinuity in Part (i). By Theorem 3.1, the solution operator depends on $u$ and its derivatives through polynomial combinations; polynomial functions are Lipschitz on bounded balls with constant $L(R) = O(R^{N-1})$ for degree-$N$ nonlinearity. By Proposition 4.3, the feature map is bi-Lipschitz, so metric entropy is preserved up to constants. Applying (Yarotsky, 2017) to the Lipschitz function $\tilde{\mathcal{G}}$ achieves rate $W^{-s/d}$. $\qquad\square$

The improvement from $(s - m)/d$ to $s/d$ is substantial. For a second-order PDE in one dimension with $s = 3$, the exponent improves from 1 to 3, meaning the same accuracy requires far fewer parameters.

## 5. The Statistical Twist: Why $\beta^* < m$

Theorem 4.4 suggests using the true PDE order $m$ as the derivative order. We now prove this is suboptimal for noisy, finite samples. This finding is our central theoretical contribution.

*Remark* 5.1 (Complementary Theoretical Perspectives). Section 4 and this section answer *different questions*. Section 4 asks: "Do derivative features help?" and answers via approximation theory. This section asks: "Given that features help, what order is optimal?" and answers via spectral bias-variance analysis. These perspectives are complementary: one justifies using derivative features at all, the other determines the best choice.

### 5.1. The Core Tension

Consider learning an operator that depends on $D^m u$ using features $D^\beta u$ with potentially different order $\beta$. Two competing effects arise:

**Bias:** If $\beta < m$, the features miss high-frequency information relevant to the target. This creates approximation error that persists regardless of sample size.

**Variance:** Differentiation amplifies noise. The Fourier multiplier $|\xi|^\beta$ grows unboundedly with frequency, meaning high-frequency noise gets amplified by factor $|\xi|^\beta$ when we compute $D^\beta u$. Higher $\beta$ means more amplification.

These effects respond oppositely to $\beta$: bias decreases while variance increases. The optimal choice balances them.

### 5.2. Spectral Analysis

To make this tradeoff precise, we assume inputs have a typical spectral structure.

**Assumption 5.2** (Spectral Decay). *The input distribution $\mu$ on $B_s(R)$ has power spectral density $S(\xi) := \mathbb{E}[|\hat{u}(\xi)|^2] = C_S|\xi|^{-2s-d}$ for $|\xi| \geq 1$, where $C_S > 0$ depends on the distribution.*

This rate is the borderline upper envelope for $H^s$ integrability: $\mathbb{E}\|u\|^2_{H^s}$ converges for any $S(\xi)$ decaying strictly faster than $|\xi|^{-2s-d}$ and is logarithmically divergent at exactly this rate. We adopt this envelope as the worst-case spectral profile on $B_s(R)$, yielding the worst-case bias integral in §5; Matérn-$s$ fields (Stein, 1999) sit at this edge. The coefficient $C_S$ cancels in the optimal $\beta^*$ formula (Theorem 5.6), so only the exponent matters.

**Proposition 5.3** (Bias-Variance Decomposition). *Under Assumption 5.2, suppose the network has effective frequency cutoff $\xi_c$ (depending on capacity $W$ but not on $\beta$). Using features $D^\beta u$ with $\beta \leq m$, the MSE decomposes as $\mathrm{MSE}(\beta) = B(\beta) + V(\beta)$ where:*

$$\text{Bias:} \quad B(\beta) = C_B \int_{|\xi| > \xi_c} |\xi|^{2(m-\beta)} S(\xi)\, d\xi, \quad (18)$$

$$\text{Variance:} \quad V(\beta) = \frac{\sigma^2 C_V}{n} \int_{|\xi| \leq \xi_{\max}} |\xi|^{2\beta}\, d\xi, \quad (19)$$

*where $\xi_{\max} = \pi/h$ is the Nyquist frequency for grid spacing $h$.*

*The bias captures the spectral mismatch: providing $D^\beta u$ instead of $D^m u$ leaves a residual $|\xi|^{m-\beta}$ that the network must learn implicitly. The variance captures noise amplification: computing $D^\beta u$ amplifies noise at frequency $\xi$ by factor $|\xi|^\beta$.*

*Both integrals admit closed forms (see Appendix E). The bias is strictly decreasing in $\beta$. The variance is strictly increasing in $\beta$ provided $\xi_{\max} > e^{1/(2\beta+d)}$, which holds for typical discretizations with $\xi_{\max} \gg 1$. There exists a unique $\beta^* \in (0, m)$ minimizing the total MSE.*

*Remark* 5.4 (Terminology). We use "bias" and "variance" by analogy with statistical learning theory. Here, $B(\beta)$

represents *approximation bias* from using derivative order $\beta < m$: high-frequency information beyond network capacity cannot be captured. Meanwhile, $V(\beta)$ represents *estimation variance* from noise amplification: computing $D^\beta u$ from noisy data amplifies observation noise by $|\xi|^\beta$ at each frequency. This mirrors the classical tradeoff where model complexity reduces bias at the cost of increased variance.

*Remark* 5.5 (Capacity Cutoff in FNO). For Fourier Neural Operators, the capacity cutoff $\xi_c = 2\pi k_{\max}/L$ is *exact*: FNO truncates to $k_{\max}$ Fourier modes by architectural design, imposing a hard frequency cutoff independent of input features. The preprocessing $D^\beta u$ changes the input's spectral content but not the architecture's representational capacity.

### 5.3. The Optimal Derivative Order

Balancing bias and variance yields our main theoretical result.

**Theorem 5.6** (Optimal Derivative Order). *Under Assumptions 4.2 and 5.2, define $\tau := 2(s-m) > 0$ and the capacity constant $A_c := 2s \ln \xi_c + (2m+d) \ln \xi_{\max}$, where $\xi_c$ is the network's frequency cutoff and $\xi_{\max} = \pi/h$ is the Nyquist frequency. The MSE-optimal derivative order satisfies:*

$$\boxed{\beta^* = m - \delta^*, \quad \delta^* = \frac{A_c - \ln(n/\sigma^2)}{2 \ln(\xi_c \cdot \xi_{\max})}} \quad (20)$$

*up to $O(1/\ln \xi_c)$ corrections.*

The formula shows that $\delta^*$ depends on the signal-to-noise ratio $n/\sigma^2$: as data quality improves, the gap shrinks and $\beta^* \to m$. The capacity constant $A_c$ captures network architecture effects. The full derivation appears in Appendix F.

*Remark* 5.7 (Data Sufficiency for Exact Physics). To leading order in $1/\ln \xi_c$, $\delta^* > 0$ strictly whenever $n/\sigma^2 < e^{A_c}$, with exact-physics recovery requiring

$$n^\star = \sigma^2 \exp(A_c). \quad (21)$$

The exponential dependence makes $n^\star$ astronomical in practice. For a representative FNO setting (unit-interval Burgers, $N=1024$, $k_{\max}=16$, giving $\xi_{\max} \approx 3200$, $\xi_c \approx 100$, $A_c \approx 68$), even at $\sigma = 10^{-3}$ this implies $n^\star \approx 10^{23}$ samples, far beyond any realistic training budget. Sub-physical regularization is therefore the operational regime at any practically achievable scale.

These results have three key implications:

1. **Strict suboptimality of $m$:** For any finite sample size $n$ and noise level $\sigma > 0$, the optimal order satisfies $\beta^* < m$ strictly. The exact PDE order is never optimal.

2. **Asymptotic recovery:** As $n \to \infty$ (or $n \geq n^\star$) or $\sigma \to 0$, the gap $\delta^* \to 0$ and $\beta^* \to m$. Physics is recovered in the large-data limit.

3. **Capacity-adaptive:** Networks with higher frequency cutoff $\xi_c$ (e.g., more Fourier modes in FNO) benefit from lower $\beta^*$, automatically trading approximation power for regularization.

# 6. $\partial$-NO: Derivative-Augmented Neural Operators

Our theoretical analysis motivates a simple architecture: neural operators with learnable fractional derivative features (Figure 1).

Given a base neural operator $\mathcal{NO}_\theta$, the $\partial$-NO augmentation replaces inputs $(x, u)$ with:

$$\partial\text{-NO}: \quad \left(x, u, s_1 \mathfrak{D}^{\beta_1} u, s_2 \mathfrak{D}^{\beta_2} u, \ldots, s_k \mathfrak{D}^{\beta_k} u\right), \quad (22)$$

where both the derivative orders $\beta_1, \ldots, \beta_k$ and the scales $s_1, \ldots, s_k$ are learnable parameters optimized jointly with network weights $\theta$. The learnable scales normalize features across different derivative orders, since $\mathfrak{D}^\beta u$ can vary dramatically in magnitude as $\beta$ changes.

While the Weyl fractional derivative (Definition 2.1) provides an elegant spectral formulation, it can introduce Fourier artifacts (Gibbs ringing) near discontinuities. We instead use the Grünwald-Letnikov (GL) finite-difference formulation:

$$\mathfrak{D}^\beta u(x) = \frac{1}{h^\beta} \sum_{j=0}^{K} (-1)^j \binom{\beta}{j} u(x - jh), \quad (23)$$

where $h$ is the grid spacing and the generalized binomial coefficients $\binom{\beta}{j}$ extend naturally to non-integer $\beta$ via the Gamma function. This formulation is differentiable with respect to $\beta$, enabling gradient-based optimization.

In practice, we omit the factor $h^{-\beta}$. For small $h$ and moderate $\beta$, this factor can be dangerously large and destabilize training. Since the learnable scales $s_k$ and the network itself absorb any constant scaling, removing $h^{-\beta}$ preserves the functional structure of the derivative while ensuring training stability. We denote backbones augmented with learnable derivative features using the $\partial$-prefix: $\partial$-**FNO**, $\partial$-**Transolver**, etc. The computational overhead is negligible.

# 7. Experiments

We validate $\partial$-NO across multiple PDEs and backbone architectures. Our experiments address two questions: Does $\partial$-augmentation improve accuracy? Does the theory correctly predict how $\beta^*$ varies with noise, sample size, and network capacity?

## 7.1. Setup

**Datasets.** We use the benchmark datasets from (Li et al., 2020): Burgers' equation (smooth, $\nu = 0.1$; near-shock, $\nu = 0.001$), Darcy flow (elliptic), and two-dimensional Navier-Stokes (incompressible turbulence at $\nu = 10^{-5}$), plus a third-order KdV dataset ($m=3$) obtained by direct numerical integration from random initial conditions (App. I).

**Backbones.** Our primary goal is to validate the theoretical predictions from Section 5. Neural operator training is particularly sensitive to hyperparameters, often requiring very small batch sizes (values of 8, 4, or even 1–2 are common (Wu et al., 2024)) to achieve good performance. This highlights the delicate nature of training such models. We use FNO (Li et al., 2020) as a canonical spectral method, its tensorized variant TFNO, the localized operator LocalNO (Liu-Schiaffini et al., 2024), and Transolver (Wu et al., 2024) as a state-of-the-art attention-based architecture, employing the THUML library (Wu et al., 2024) with default hyperparameters.

**Derivative features.** For the second-order benchmarks (Burgers, Darcy, NS), $m = 2$ and we use two learnable derivative features. For 1D problems: $\mathfrak{D}_x^{\beta_1} u$ and $\mathfrak{D}_x^{\beta_2} u$, initialized at $\beta_1 = 1.0$ and $\beta_2 = 2.0$; for the third-order KdV ($m = 3$) we use a third feature $\mathfrak{D}_x^{\beta_3} u$ initialized at $\beta_3 = 3.0$. For 2D problems, we include directional derivatives $\mathfrak{D}_x^{\beta_{x,1}} u, \mathfrak{D}_x^{\beta_{x,2}} u, \mathfrak{D}_y^{\beta_{y,1}} u, \mathfrak{D}_y^{\beta_{y,2}} u$ and one mixed partial $\mathfrak{D}_y^{\beta_{y,c}} \mathfrak{D}_x^{\beta_{x,c}} u$. Each order is learned independently. In particular, the mixed partial has two learnable orders $(\beta_{x,c}, \beta_{y,c})$ and a single shared scaling factor.

**Training.** All models train with AdamW. The $\beta$ parameters use $10\times$ higher learning rate ($10^{-2}$ vs $10^{-3}$) because they sit at the input level: gradients through them propagate back through the full backbone and attenuate accordingly. We report relative $L^2$ errors (%). Full hyperparameters and implementation details are provided in Appendix I.

## 7.2. Validation of $\partial$-NO Improvements

Table 1 summarizes our main results: five PDE benchmarks, four backbones, and both clean ($\sigma=0$) and noisy ($\sigma=0.05$) regimes. The $\partial$-augmentation improves every backbone/dataset/noise combination tested; the separate CNO benchmarks (§7.4, Table 2) show even larger gains. A broader stress test under Gaussian and uniform additive noise with $\sigma \in [0.05, 0.3]$ confirms this robustness across every tested setting (Appendix L.3), and a matched-parameter control isolates the gains to the derivative inductive bias rather than added capacity (Appendix L.2). Further robustness analyses (emergent directional isotropy on 2D/3D isotropic PDEs in Figs. 6, 7; resolution-dependent gains in Fig. 4; complementarity with PINO physics-informed training in Table 9) are reported in the appendix.

*Table 1.* Relative $L^2$ error (%), averaged over 5 seeds, across 5 PDE benchmarks $\times$ 4 backbones $\times$ {clean, noisy}. Bold = best per (backbone, dataset, noise) cell. $\partial$-augmentation improves every cell tested.

| | **Burgers** ($\nu$=0.1) | | **Burgers** ($\nu$=0.001) | | **KdV** | | **Darcy** (Elliptic) | | **NS2D** ($\nu$=$10^{-5}$) | |
|---|---|---|---|---|---|---|---|---|---|---|
| | $\sigma$=0 | $\sigma$=0.05 | $\sigma$=0 | $\sigma$=0.05 | $\sigma$=0 | $\sigma$=0.05 | $\sigma$=0 | $\sigma$=0.05 | $\sigma$=0 | $\sigma$=0.05 |
| FNO | 0.065 | 0.250 | 1.138 | 1.632 | 1.598 | 2.255 | 5.643 | 5.590 | 3.479 | 3.575 |
| $\partial$-FNO | **0.055** | **0.190** | **0.837** | **1.366** | **1.549** | **2.195** | **5.426** | **5.415** | **3.428** | **3.511** |
| *Improv.* | 15.1% | 24.1% | 26.5% | 16.3% | 3.0% | 2.7% | 3.8% | 3.1% | 1.5% | 1.8% |
| TFNO | 0.071 | 0.209 | 0.702 | 1.032 | 2.312 | 2.745 | 5.223 | 5.206 | 3.428 | 3.534 |
| $\partial$-TFNO | **0.062** | **0.178** | **0.676** | **0.917** | **2.227** | **2.660** | **4.843** | **4.906** | **3.228** | **3.368** |
| *Improv.* | 11.9% | 14.7% | 3.7% | 11.2% | 3.7% | 3.1% | 7.3% | 5.8% | 5.8% | 4.7% |
| LocalNO | 0.063 | 0.236 | 0.768 | 1.272 | 2.587 | 3.625 | 5.511 | 5.534 | 1.994 | 2.620 |
| $\partial$-LocalNO | **0.060** | **0.199** | **0.732** | **1.203** | **2.352** | **3.441** | **5.430** | **5.487** | **1.984** | **2.533** |
| *Improv.* | 6.1% | 15.8% | 4.7% | 5.4% | 9.1% | 5.1% | 1.5% | 0.9% | 0.5% | 3.3% |
| Transolver | 0.990 | 0.765 | 18.988 | 7.521 | 22.427 | 16.334 | 14.645 | 15.332 | 4.301 | 4.821 |
| $\partial$-Transolver | **0.709** | **0.736** | **4.633** | **4.049** | **17.957** | **9.588** | **13.125** | **11.212** | **4.043** | **4.341** |
| *Improv.* | 28.4% | 3.8% | 75.6% | 46.2% | 19.9% | 41.3% | 10.4% | 26.9% | 6.0% | 9.9% |

## 7.3. Theory Validation

Theorem 5.6 makes testable predictions about how $\beta^*$ responds to noise $\sigma$, sample size $n$, and network capacity (Fourier modes $k$). Since $\xi_c = 2\pi k$, the formula simplifies to show explicit dependence on the three experimental parameters:

$$\delta^* \propto \frac{2s \ln k + 2 \ln \sigma - \ln n + C}{2 \ln k + C'} \qquad (24)$$

where $C, C'$ are problem-dependent constants. This yields three monotonicities under Assumption 4.2: $\sigma \uparrow \Rightarrow \beta^* \downarrow$, $n \uparrow \Rightarrow \beta^* \uparrow$, and $k \uparrow \Rightarrow \beta^* \downarrow$. Near the smoothness boundary $s \approx m + d/2$, the capacity-effect margin shrinks and the empirical $k$-sweep weakens, as detailed below. We test all three on Burgers (smooth and shock), using a single learnable $\beta$ and varying one factor at a time.

Figures 2 and 3 confirm both unconditional predictions on smooth and shock Burgers, and the capacity effect (a decrease of $\beta^*$ with $k$ in both regimes, attenuating near the smoothness boundary):

**Noise** ($\sigma \uparrow \Rightarrow \beta^* \downarrow$)**.** The $+2 \ln \sigma$ term in (24) predicts that increasing noise increases $\delta^*$. Experimentally, learned $\beta^*$ decreases monotonically with $\sigma$ on both smooth and shock Burgers, exactly as predicted. The network automatically regularizes by learning lower derivative orders when data is noisy.

**Sample size** ($n \uparrow \Rightarrow \beta^* \uparrow$)**.** The $-\ln n$ term predicts that more data decreases $\delta^*$ and drives $\beta^* \to m$. Experimentally, learned $\beta^*$ increases monotonically with $n$, approaching but not reaching $m = 2$ at any practically achievable $n$. The persistent gap $\delta^* > 0$ is consistent with Remark 5.7: the recovery threshold $n^* = \sigma^2 \exp(A_c)$ is far beyond sample sizes accessible in this sweep.

**Capacity** ($k \uparrow \Rightarrow \beta^* \downarrow$)**.** The $+2s \ln k$ term in the numera-

tor competes with $+2 \ln k$ in the denominator. Under Assumption 4.2 ($s > m + d/2$), the numerator dominates and $k \uparrow \Rightarrow \delta^* \uparrow$; near the smoothness boundary the margin shrinks and the empirical effect weakens. This predicts qualitatively different behavior for smooth vs. shock Burgers. Experimentally, we observe two qualitatively consistent but differently scaled behaviors in Figures 2 and 3: smooth Burgers ($\tau \approx 2.2$) shows a clean monotonic decrease, while shock Burgers ($\tau \approx 0.8$) shows a milder, noisier decrease over the same range. The consistency across smooth and shock regimes supports the bias-variance framework.

## 7.4. Differential Inductive Bias: $\partial$-CNO

A natural question is whether $\partial$-NO's gains come from an increased local processing capacity or from the specific inductive bias from fractional differentiation. Convolutional architectures provide a sharp test: a generic convolution *can* represent a finite-difference derivative, but it is not *constrained* to learn one. To validate the importance of the $\partial$-NO's inductive bias, we instantiate $\partial$-augmentation on the Convolutional Neural Operator (Raonic et al., 2023) (CNO) and compare $\partial$-CNO against the vanilla CNO across the seven PDE benchmarks of Raonic et al. (2023).

Results are in Table 2. $\partial$-CNO improves CNO in all settings, with peak gains up to **55–65%** on Poisson and Wave, and $\sim 39\%$ on NS Shear. The pattern is consistent with the differential interpretation: gains are largest on PDEs whose solution operators most depend on global differential structure (elliptic Poisson, shear-dominated NS, wave propagation), and smaller on data-dominated settings (Darcy, Airfoil). Because CNO is built entirely from learnable convolutions, the persistent improvement from imposing the GL derivative structure on the input is direct evidence that derivative features carry information convolutional layers do not reliably acquire from data alone.

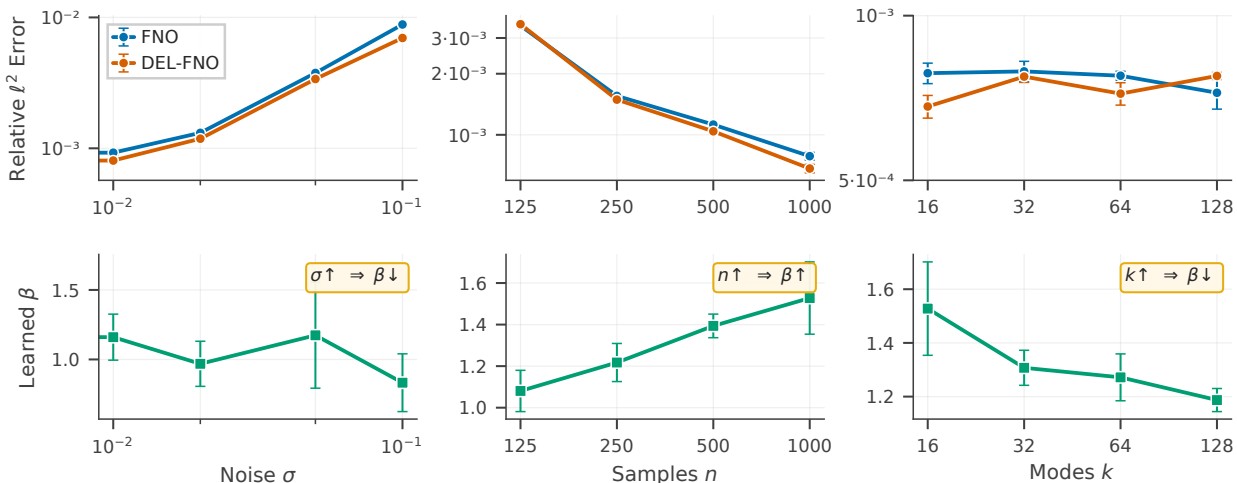

*Figure 2.* **Burgers (smooth, $\nu=0.1$): Theory vs. experiment.** Top: relative $L^2$ error; bottom: learned $\beta^*$. Theorem 5.6's two unconditional predictions are confirmed: (left) $\sigma \uparrow \Rightarrow \beta^* \downarrow$; (center) $n \uparrow \Rightarrow \beta^* \uparrow$, with $\beta^*$ approaching but not reaching $m=2$, consistent with the unreachable recovery threshold (Remark 5.7). At smoothness $\tau \approx 2.2$ (comfortably above the Assumption 4.2 threshold $s > m + d/2$), the capacity formula predicts $k \uparrow \Rightarrow \beta^* \downarrow$ (right), which holds monotonically.

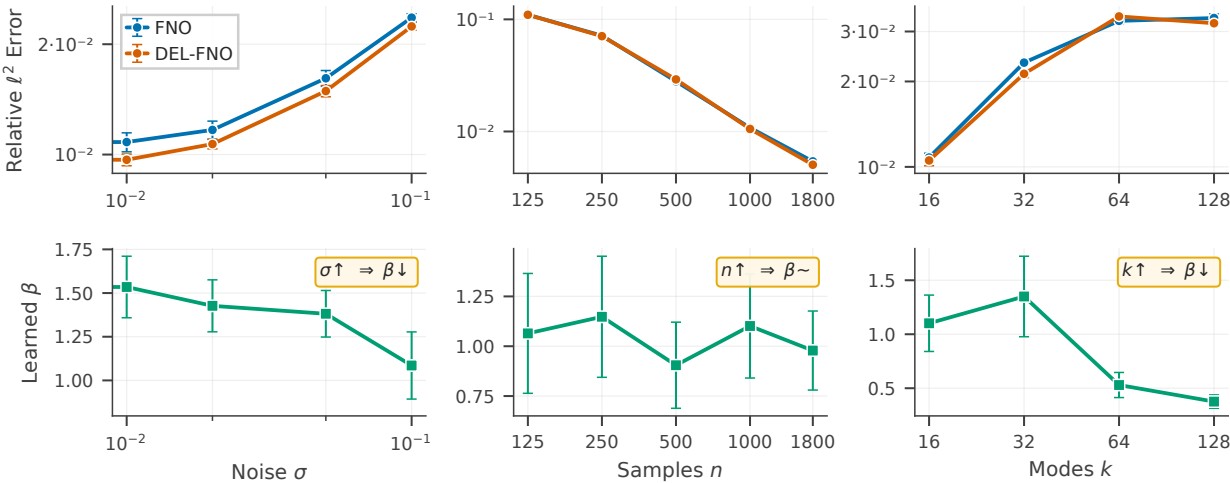

*Figure 3.* **Burgers (near-shock, $\nu=0.001$): Theory vs. experiment.** Same panels as Figure 2. The noise prediction ($\sigma \uparrow \Rightarrow \beta^* \downarrow$) holds clearly; the sample-size effect ($n \uparrow \Rightarrow \beta^* \uparrow$) attenuates to a noisy plateau near the smoothness boundary ($\tau \approx 0.8$), as expected when Assumption 4.2 is marginal. At rough-data smoothness $\tau \approx 0.8$ (near the Assumption 4.2 smoothness boundary), the capacity formula admits a weakened $k$-effect; the empirical $k$-sweep shows the predicted attenuation: a milder, noisier decrease in $\beta^*$ rather than the clean monotone of the smooth case, consistent with this regime change.

*Table 2.* $\partial$-CNO vs. CNO on seven benchmarks of Raonic et al. (2023), evaluated under both relative $L^2$ error and median relative $L^1$ error (averaged over 5 seeds). Bold = best per (dataset, noise, metric) cell. $\partial$-CNO improves CNO in **14/14** settings under *both* metrics.

|  |  | **Poisson** | | **NS Shear** | | **Wave** | | **Allen–Cahn** | | **Transp. Cont.** | | **Darcy** | | **Airfoil** | |
|---|---|---|---|---|---|---|---|---|---|---|---|---|---|---|---|
|  | $\sigma$ | 0 | 0.05 | 0 | 0.05 | 0 | 0.05 | 0 | 0.05 | 0 | 0.05 | 0 | 0.05 | 0 | 0.05 |
| $L^2$ | CNO | 1.788 | 3.664 | 9.686 | 12.218 | 0.590 | 1.990 | 2.618 | 1.980 | 0.416 | 0.413 | 0.763 | 0.724 | 1.184 | 1.130 |
|  | $\partial$-CNO | **1.053** | **1.293** | **7.104** | **7.746** | **0.520** | **0.900** | **1.632** | **1.575** | **0.353** | **0.381** | **0.752** | **0.718** | **1.137** | **1.111** |
|  | *Improv.* | 41.2% | 64.7% | 26.7% | 36.6% | 11.8% | 54.8% | 37.7% | 20.4% | 15.2% | 7.7% | 1.4% | 0.8% | 4.0% | 1.7% |
| $L^1$ | CNO | 1.200 | 2.504 | 8.120 | 10.080 | 0.440 | 1.550 | 0.870 | 0.940 | 0.358 | 0.380 | 0.514 | 0.482 | 0.446 | 0.473 |
|  | $\partial$-CNO | **0.660** | **0.884** | **5.910** | **6.110** | **0.439** | **0.692** | **0.646** | **0.857** | **0.341** | **0.352** | **0.501** | **0.457** | **0.443** | **0.410** |
|  | *Improv.* | 45.0% | 64.7% | 27.2% | 39.4% | 0.2% | 55.4% | 25.7% | 8.8% | 4.7% | 7.5% | 2.6% | 5.1% | 0.7% | 13.3% |

# 8. Related Work

**Neural operators.** The field has grown rapidly since FNO (Li et al., 2020) and DeepONet (Lu et al., 2021). Recent work explores attention mechanisms (Cao, 2021; Hao et al., 2023; Wu et al., 2024), wavelets (Tripura & Chakraborty, 2023), and implicit representations (Serrano et al., 2023). Theoretical foundations include approximation bounds (Kovachki et al., 2023) and universal approximation results. Our ∂-augmentation is orthogonal to these advances: it wraps any backbone with learnable derivative features, requiring no architectural modification.

**Local-feature operators.** A parallel line of work injects locality into neural operators: Conv-FNO (Liu et al., 2025) attaches local convolutions to FNO, LocalFNO (Liu-Schiaffini et al., 2024) introduces localized integral and differential kernels, and CNO (Raonic et al., 2023) replaces spectral layers with structured convolutions throughout. ∂-NO is complementary: it provides a specific local operation (fractional differentiation) with a built-in inductive bias for PDE structure. Our ∂-CNO experiments (§7) directly isolate this bias from generic local processing.

**Derivative and physics-aware features.** Recent work has explored providing derivative information to neural operators. Equation-aware emulators (Zhu et al., 2025) encode full PDE structure including nonlinear terms, achieving improvements but requiring the PDE form a priori. Physics-informed approaches like PINNs (Raissi et al., 2019) and the physics-informed neural operator (Li et al., 2024) incorporate derivatives through loss terms via automatic differentiation. Our approach differs in three ways: (1) we learn derivative orders rather than specifying them from known physics; (2) we provide derivatives as input features, computed via FFT or GL; and (3) our bias-variance analysis reveals that optimal orders should be less than the PDE order, a phenomenon that equation-matching approaches do not address.

**Spectral perspectives.** FNO performs convolutions in Fourier space with a hard mode truncation at $k_{\max}$. Our analysis exploits this structure: the capacity cutoff $\xi_c$ in Theorem 5.6 is exact for FNO and other neural operators with a fixed number of Fourier modes, making the bias-variance tradeoff particularly sharp. More broadly, spectral bias in neural networks, where low frequencies are learned before high frequencies (Rahaman et al., 2019), suggests that derivative features may help by preprocessing high-frequency content into a form the network can access earlier in training.

**Fractional calculus.** Fractional derivatives model anomalous diffusion, memory effects, and non-local phenomena (Metzler & Klafter, 2000), and have found applications in financial machine learning for constructing stationary features that preserve memory (López de Prado, 2018). We use them differently: as a continuous parameterization enabling gradient-based optimization over derivative order. This treats the fractional derivative as a learnable feature transformation, not as a physical model of sub-diffusion or long-range dependence. Connections between learned orders and anomalous diffusion remain an interesting direction for future work.

# 9. Discussion

We introduced ∂-NO, a simple augmentation providing learnable fractional derivative features to any neural operator backbone. Through Picard iteration on mild solutions, we showed how derivatives of the initial condition directly contribute to the solution operator, providing theoretical grounding for derivative features. Providing these features improves approximation rates from $O(W^{-(s-m)/d})$ to $O(W^{-s/d})$, a substantial gain in parameter efficiency.

Our central finding is that the optimal derivative order $\beta^*$ is less than the PDE order $m$, with the gap characterized by a spectral bias-variance tradeoff. The closed-form expression for $\delta^* = m - \beta^*$ correctly predicts how the optimal order responds to noise, sample size, and capacity, as confirmed experimentally. For FNO and other operators with fixed Fourier modes, the capacity cutoff is exact. Learning $\beta$ achieves automatic spectral regularization: the network selects $\beta^*$ in response to noise, sample size, and capacity, as predicted by Theorem 5.6 and confirmed empirically.

**Limitations.** Four scope boundaries: (i) the Grünwald-Letnikov stencil assumes a regular grid (extensions via graph fractional Laplacians or RBF operators are open); (ii) Theorem 5.6's sharp predictions require $s > m + d/2$ and degrade to a directional guide near shocks, as the near-shock Burgers experiments empirically confirm; (iii) we preprocess only spatial inputs, leaving temporal fractional derivatives as an orthogonal direction; and (iv) the theorem characterizes the population-optimal $\beta^*$ rather than the optimizer's converged value. Empirically, $\beta$ stabilizes within 200–400 AdamW epochs across every backbone and PDE we tested, but formal trajectory guarantees remain open.

Stepping back from these scope considerations, the central finding $\beta^* < m$ illustrates a broader principle for physics-informed machine learning at finite scales: inductive biases derived from governing equations should be *parameterized rather than prescribed*, allowing the optimizer to locate its own compromise between approximation power and estimation variance. Under finite, noisy data, the statistically optimal physics is rarely the literal equation; learning the inductive bias is what turns physical knowledge into statistical efficiency.

## Impact Statement

This work advances methodology for scientific machine learning, with potential applications in a wide range of scientific fields. The finding that learned derivative orders should be sub-physical has implications for how we incorporate domain knowledge into machine learning systems: matching physics exactly may not always be optimal. We identify no specific ethical concerns.

## Code Availability

The code for $\partial$-NO is available at https://github.com/FaresBMehouachi/delNO. The authors declare no competing interests.

## Acknowledgments

This work was supported by the NYUAD Center for Interacting Urban Networks (CITIES), funded by Tamkeen under the NYUAD Research Institute Award CG001. The views expressed in this article are those of the authors and do not reflect the opinions of CITIES or their funding agencies.

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

# A. Table of Notation

*Table 3.* Summary of notation used throughout the paper.

| Notation | Meaning |
|---|---|
| *Spaces and functions* | |
| $\Omega \subset \mathbb{R}^d$ | Spatial domain |
| $d$ | Spatial dimension |
| $H^s(\Omega)$ | Sobolev space of regularity $s$ |
| $B_s(R)$ | Sobolev ball $\{u \in H^s : \|u\|_{H^s} \leq R\}$ |
| $u, u_0$ | Input function / initial condition |
| $v$ | Solution function |
| *PDE structure* | |
| $m$ | Differential order of the PDE |
| $\mathcal{L}$ | Linear differential operator |
| $\mathcal{N}$ | Nonlinearity |
| $N$ | Degree of polynomial nonlinearity |
| $G_t$ | Green's function / semigroup $e^{t\mathcal{L}}$ |
| $\nu$ | Viscosity coefficient |
| *Operators and features* | |
| $\mathcal{G}, \mathcal{G}^\dagger$ | Solution operator (learned / true) |
| $\mathcal{G}_\theta$ | Neural operator with parameters $\theta$ |
| $\Phi^{[m]}$ | Feature map $(u, D^1 u, \ldots, D^m u)$ |
| $D^\beta$ | Fractional derivative of order $\beta$ |
| $\mathfrak{D}^\beta$ | GL approximation (Alternatively Weyl via FFT on periodic domains) |
| *Derivative orders* | |
| $\beta$ | Learnable derivative order |
| $\beta^*$ | Optimal derivative order (MSE-minimizing) |
| $\delta^* = m - \beta^*$ | Gap from PDE order |
| $s_k$ | Learnable scale for $k$-th derivative feature |
| *Spectral quantities* | |
| $\xi$ | Frequency variable |
| $\hat{u}(\xi) = \mathcal{F}[u]$ | Fourier transform of $u$ |
| $S(\xi)$ | Power spectral density $\mathbb{E}[|\hat{u}(\xi)|^2]$ |
| $\xi_c$ | Network capacity cutoff frequency |
| $\xi_{\max} = \pi/h$ | Nyquist frequency |
| $k, k_{\max}$ | Fourier modes / maximum modes |
| *Statistical quantities* | |
| $n$ | Number of training samples |
| $\sigma^2$ | Observation noise variance |
| MSE, $B, V$ | Mean squared error, bias, variance |
| $\tau = 2(s - m)$ | Regularity gap parameter |
| *Implementation* | |
| $W$ | Network width / number of parameters |
| $h$ | Grid spacing |
| $K$ | GL kernel size (number of terms) |
| $w_k^{(\beta)}$ | GL weights for order $\beta$ |
| $\psi = \Gamma'/\Gamma$ | Digamma function |

## B. Proof of Theorem 3.1 (Derivative Emergence)

We prove that derivatives of the initial condition naturally enter the solution operator through Picard iteration.

*Proof.* Consider the semilinear PDE:

$$\partial_t u = \mathcal{L}u + \mathcal{N}[u], \quad u(0) = u_0, \tag{25}$$

where $\mathcal{L}$ is a linear differential operator of order $m$ and $\mathcal{N}$ is a polynomial nonlinearity of degree $N$.

**Step 1: Mild solution formulation.** The mild solution satisfies the integral equation:

$$u(t) = G_t * u_0 + \int_0^t G_{t-s} * \mathcal{N}[u(s)] \, ds, \tag{26}$$

where $G_t = e^{t\mathcal{L}}$ is the semigroup generated by $\mathcal{L}$. For the heat operator $\mathcal{L} = \nu\Delta$, this is the heat kernel. For general $\mathcal{L}$, it is the Green's function.

**Step 2: Picard iteration.** Define iterates:

$$u^{(0)}(t) = G_t * u_0, \tag{27}$$

$$u^{(k+1)}(t) = G_t * u_0 + \int_0^t G_{t-s} * \mathcal{N}[u^{(k)}(s)] \, ds. \tag{28}$$

Under standard assumptions (bounded $u_0$, Lipschitz $\mathcal{N}$, finite time horizon), the Banach fixed-point theorem guarantees convergence $u^{(k)} \to u$ in an appropriate function space.

**Step 3: Expansion of the semigroup.** The semigroup satisfies $\partial_t G_t = \mathcal{L}G_t$ with $G_0 = I$. For analytic semigroups (which include all parabolic operators with constant coefficients), Taylor expanding in $s$:

$$G_s * u_0 = u_0 + s \cdot \mathcal{L}u_0 + \frac{s^2}{2} \cdot \mathcal{L}^2 u_0 + O(s^3). \tag{29}$$

The term $\mathcal{L}u_0$ involves $m$-th order derivatives of $u_0$ (e.g., $\mathcal{L}u_0 = \nu\Delta u_0$ for diffusion). The $O(s^3)$ remainder is uniform over $u_0 \in B_s(R)$ by standard semigroup estimates.

**Step 4: Expansion of the nonlinearity.** Since $\mathcal{N}$ is Fréchet differentiable:

$$\mathcal{N}[G_s * u_0] = \mathcal{N}[u_0 + s\mathcal{L}u_0 + O(s^2)] = \mathcal{N}[u_0] + s \cdot D\mathcal{N}[u_0][\mathcal{L}u_0] + O(s^2), \tag{30}$$

where $D\mathcal{N}[u_0][v]$ is the directional derivative of $\mathcal{N}$ at $u_0$ in direction $v$.

For polynomial $\mathcal{N}$, this derivative has explicit form. For example, if $\mathcal{N}[u] = -u\partial_x u$, then:

$$D\mathcal{N}[u_0][v] = -v\partial_x u_0 - u_0\partial_x v. \tag{31}$$

When $v = \mathcal{L}u_0 = \nu\partial_{xx}u_0$, we get terms involving $\partial_{xx}u_0 \cdot \partial_x u_0$ and $u_0 \cdot \partial_{xxx}u_0$.

**Step 5: First Picard iterate.** The first iterate at final time $T$ is:

$$u^{(1)}(T) = G_T * u_0 + \int_0^T G_{T-s} * \mathcal{N}[G_s * u_0] \, ds. \tag{32}$$

Substituting the expansions:

$$\int_0^T G_{T-s} * \mathcal{N}[G_s * u_0] \, ds = \int_0^T G_{T-s} * \left(\mathcal{N}[u_0] + s \cdot D\mathcal{N}[u_0][\mathcal{L}u_0] + O(s^2)\right) ds \tag{33}$$

$$= \mathcal{N}[u_0] * \int_0^T G_{T-s} \, ds + D\mathcal{N}[u_0][\mathcal{L}u_0] * \int_0^T s \cdot G_{T-s} \, ds + O(T^3), \tag{34}$$

where we exchanged the order of convolution and integration by Fubini's theorem (valid since $G_{T-s}$ is integrable on $[0,T]$ for the semigroups considered).

**Step 6: Averaged Green's operators.** Define:

$$\bar{G}_T := \frac{1}{T} \int_0^T G_{T-s} \, ds, \tag{35}$$

$$\tilde{G}_T := \frac{2}{T^2} \int_0^T s \cdot G_{T-s} \, ds. \tag{36}$$

These are bounded linear operators for $T > 0$ (the semigroup is smoothing).

**Step 7: Structure of the solution operator.** Combining:

$$u^{(1)}(T) = G_T * u_0 + T \cdot \bar{G}_T * \mathcal{N}[u_0] + \frac{T^2}{2} \cdot \tilde{G}_T * D\mathcal{N}[u_0][\mathcal{L}u_0] + O(T^3). \tag{37}$$

The true solution $u(T) = \lim_{k \to \infty} u^{(k)}(T)$ differs from $u^{(1)}(T)$ by higher-order Picard corrections. For fixed $T < T_*(R)$, these corrections are bounded perturbations that preserve the dependence structure. Defining the remainder $\mathcal{R}(u_0, T) := \mathcal{G}[u_0] - G_T * u_0 - T \cdot \bar{G}_T * \mathcal{N}[u_0]$, we have $\|\mathcal{R}(u_0, T)\|_{L^2} \leq C(R)T^2$ by standard Picard remainder estimates. The constant $C(R)$ depends polynomially on $R$ through the Lipschitz constant of $\mathcal{N}$ on $B_s(R)$ and the semigroup bounds.

**Step 8: Identifying the features.** The solution depends on $u_0$ through:

- $u_0$ (zeroth order): appears in $G_T * u_0$ and in $\mathcal{N}[u_0]$

- $\mathcal{L}u_0$ (order $m$): appears in the expansion of $G_s * u_0$

- $\partial^j u_0$ for $j \leq m$: appear in $\mathcal{N}[u_0]$ if $\mathcal{N}$ involves derivatives

- $u_0^n$ for $n \leq N$: appear in $\mathcal{N}[u_0]$ if $\mathcal{N}$ is polynomial degree $N$

This completes the proof that the solution operator $\mathcal{G}[u_0] = u(T)$ depends on derivatives of $u_0$ up to order $m$ and powers up to $u_0^N$. □

### B.1. Explicit Features for Specific PDEs

**Burgers equation.** $\partial_t u + u\partial_x u = \nu\partial_{xx} u$, so $\mathcal{L} = \nu\partial_{xx}$ and $\mathcal{N}[u] = -u\partial_x u$.

Features: $\{u_0, \partial_x u_0, \partial_{xx} u_0, u_0\partial_x u_0\}$.

**Navier-Stokes (2D vorticity).** $\partial_t \omega + \mathbf{u} \cdot \nabla\omega = \nu\Delta\omega$, where $\mathbf{u} = \nabla^\perp\psi$ and $\Delta\psi = -\omega$.

Features: $\{\omega_0, \nabla\omega_0, \Delta\omega_0, \mathbf{u}_0 \cdot \nabla\omega_0\}$.

**Darcy flow.** $-\nabla \cdot (a\nabla p) = f$. This is elliptic; the solution operator maps coefficient $a \mapsto$ pressure $p$.

Features: $\{a, \nabla a, \nabla^2 a\}$ (encoding coefficient variations).

**KdV.** $\partial_t u + u\,\partial_x u + \delta\,\partial_{xxx} u = 0$ with $\delta = 2.5 \times 10^{-3}$ (App. I), so $\mathcal{L} = -\delta\,\partial_{xxx}$ ($m = 3$) and $\mathcal{N}[u] = -u\,\partial_x u$. The dependence on $\partial_{xxx} u_0$, and hence the feature set $\{u_0, \partial_x u_0, \partial_{xx} u_0, \partial_{xxx} u_0, u_0\,\partial_x u_0\}$, is independent of the value of $\delta$.

### B.2. Extension to Arbitrary Times and Stationary Problems

Theorem 3.1 establishes derivative dependence for $T < T_*(R)$. We now show this structure persists for arbitrary times and extends to stationary problems.

**Iterated Picard for large $T$.** The key observation is simple: once derivatives $\{D^j u_0\}_{j \leq m}$ appear in the first Picard iterate, they propagate through all subsequent iterations. For $T > T_*(R)$, the solution $u(T)$ can be written as a (possibly complex) functional of $\{u_0, Du_0, \ldots, D^m u_0\}$. The specific form of this functional (involving compositions of semigroups, integrals, and nonlinear operations) is precisely what the neural operator learns. Our contribution is showing that providing $\{D^j u_0\}$ as input features gives the network direct access to the quantities it needs, rather than requiring it to learn differentiation implicitly.

**Stationary (elliptic) problems.** For stationary PDEs like Darcy flow $-\nabla \cdot (a\nabla p) = f$, there is no time evolution. The solution operator $\mathcal{G} : a \mapsto p$ depends directly on derivatives of the coefficient through the elliptic structure.

Expanding around a reference coefficient $a_0$:

$$p[a] = p[a_0] + \mathcal{L}^{-1}[\nabla \cdot (\delta a \nabla p[a_0])] + O(\|\delta a\|^2), \tag{38}$$

where $\mathcal{L} = -\nabla \cdot (a_0 \nabla \cdot)$ and $\delta a = a - a_0$. The linear response involves $\nabla(\delta a)$, so the solution operator depends on $\{a, \nabla a\}$. Higher-order terms introduce $\nabla^2 a$, yielding the feature set $\{a, \nabla a, \nabla^2 a\}$ for this second-order operator.

The key insight is identical to the parabolic case: the PDE structure induces dependence on derivatives up to the operator order $m$, and providing these as features aids learning.

**Quasilinear and non-polynomial nonlinearities.** The analysis extends beyond strictly polynomial $\mathcal{N}$. For quasilinear PDEs like Navier-Stokes, where the velocity $\mathbf{u} = \nabla^{\perp}\Delta^{-1}\omega$ involves inverting the Laplacian, the key observation is that $\Delta^{-1}$ is a *bounded* operator on Sobolev spaces; it does not introduce new dependence on derivatives of $u_0$. What matters is that the PDE structure (through $\mathcal{L}$ and $\mathcal{N}$) creates dependence on $\{u_0, D^j u_0\}_{j \leq m}$; other operations (like $\Delta^{-1}$) are learned by the neural operator. This justifies applying our framework to Navier-Stokes (with vorticity input $\omega$) as a second-order problem.

### B.3. General Evolution Equations: A Spectral Argument

The Picard analysis of §B establishes derivative dependence under the parabolic hypotheses of Theorem 3.1 (semilinear evolution with semigroup-generating $\mathcal{L}$ and polynomial $\mathcal{N}$). Here we present a complementary spectral argument that extends the same derivative structure to general evolution equations, including hyperbolic and fractional-in-time PDEs. This addresses the role of the temporal order in determining the derivative-feature structure.

**Setup.** Consider the abstract evolution problem

$$\partial_t^a u = \mathcal{L}u, \quad u(\cdot, 0) = u_0, \quad t \in [0, T], \tag{39}$$

where $\partial_t^a$ is the Caputo fractional derivative of order $a > 0$ and $\mathcal{L}$ is a linear operator whose Fourier symbol $P(\xi)$ satisfies $P(\xi) \asymp |\xi|^m$ as $|\xi| \to \infty$. For $a > 1$, the equation requires $\lceil a \rceil$ initial conditions; we set the higher-order ones to zero for clarity (they enter linearly and admit the same analysis). Assume zero-mean initial data $\hat{u}_0(0) = 0$, standard for periodic domains and PDEs with vanishing constants.

**Step 1: Fourier reduction.** The spatial Fourier transform reduces (39) to a family of scalar fractional ODEs indexed by $\xi$:

$$\partial_t^a \hat{u}(t, \xi) = P(\xi)\,\hat{u}(t, \xi), \quad \hat{u}(0, \xi) = \hat{u}_0(\xi). \tag{40}$$

**Step 2: Laplace transform and resolvent.** Applying the Laplace transform in $t$ and solving algebraically yields

$$\hat{u}(T, \xi) = R_a(P(\xi), T) \cdot \hat{u}_0(\xi), \tag{41}$$

where the resolvent multiplier $R_a(z, T)$ is the inverse Laplace transform of $s^{a-1}/(s^a - z)$. Three canonical cases:

- Parabolic ($a = 1$): $R_1(z, T) = e^{zT}$ (heat-type semigroup).
- Hyperbolic ($a = 2$): $R_2(z, T) = \cos(\sqrt{-z}\,T)$ (wave-type oscillator).
- Fractional ($0 < a < 2$, $a \neq 1$): $R_a(z, T) = E_a(zT^a)$, with $E_a$ the Mittag-Leffler function.

**Step 3: Factorization through $|\xi|^{-m}$.** Since $P(\xi) = \Theta(|\xi|^m)$, the resolvent $R_a$ acts as an order-$m$ multiplier in frequency space. The zero-mean condition $\hat{u}_0(0) = 0$ ensures $|\xi| > 0$ on the support of $\hat{u}_0$, so we may factor

$$R_a(P(\xi), T)\,\hat{u}_0(\xi) = \underbrace{\frac{R_a(P(\xi), T)}{|\xi|^m}}_{=:Q_a(\xi, T)} \cdot \underbrace{|\xi|^m \hat{u}_0(\xi)}_{= \widehat{D^m u_0}(\xi)}. \tag{42}$$

The first factor $Q_a(\xi, T)$ is a zeroth-order Fourier multiplier, bounded uniformly in $\xi$ for fixed $T$. The second factor is the Fourier transform of $D^m u_0$. Inverting the transform gives

$$\mathcal{G}[u_0] \;=\; u(\cdot, T) \;=\; Q_a(T) * D^m u_0, \tag{43}$$

where $Q_a(T)$ is a bounded linear operator on $L^2(\Omega)$.

**Step 4: Consequence for derivative features.**    Providing $D^b u_0$ as an input feature for any $b \leq m$ reduces the residual differentiation burden from order $m$ to order $m - b$:

$$\mathcal{G}[u_0] \;=\; Q_a(T) * D^{m-b}(D^b u_0). \tag{44}$$

This is the spectral analogue of the Picard derivative structure (Theorem 3.1). It holds for every temporal order $a > 0$ for which $R_a$ is well-defined, with the caveat that the operator norm $\|Q_a(T)\|$ depends on $a$ and $T$ but *not* on the choice of input order $b$.

**Nonlinear extension.**    For semilinear PDEs $\partial_t^a u = \mathcal{L}u + \mathcal{N}[u]$, linearizing around the linear flow $u^{\mathrm{lin}}(t) = R_a(\mathcal{L}, t)u_0$ and applying Duhamel's principle yields

$$u(T) = R_a(\mathcal{L}, T)u_0 + \int_0^T R_a(\mathcal{L}, T - s)\,\mathcal{N}[u^{\mathrm{lin}}(s)]\,ds + \mathcal{R}(u_0, T), \tag{45}$$

with $\|\mathcal{R}\|_{L^2} \leq C(R)T^{2a}$ uniformly on $B_s(R)$ under Assumption 4.2 and the hypotheses of Theorem 3.1. The Fréchet derivative of $\mathcal{G}$ satisfies a linearized equation with the same resolvent structure, so the factorization through $|\xi|^{-m}$ persists term by term.

**Role of the temporal order.**    The temporal order $a$ enters through three quantities: (i) the number of initial conditions $\lceil a \rceil$ (information at $t = 0$); (ii) the specific functional form of $R_a$ (heat for $a = 1$, wave for $a = 2$, Mittag-Leffler for $0 < a < 2$); and (iii) the rate at which $\|Q_a(T)\|$ decays in $T$. Crucially, $a$ does not change the derivative-feature structure, which is determined entirely by the spatial order $m$ via $P(\xi) \asymp |\xi|^m$. This is why our framework, calibrated on parabolic $(a = 1)$ Burgers and elliptic Darcy, transfers to other temporal regimes provided the spatial order $m$ is matched.

## C. Proof of Proposition 4.3: Bi-Lipschitz Stability

*Proof.* Let $\Phi^{[m]}[u] = (u, D^1 u, \ldots, D^m u)$ map $H^s(\Omega) \to \mathcal{X} = H^s \times H^{s-1} \times \cdots \times H^{s-m}$.

**Lower bound (information preservation).** Since $\Phi^{[m]}$ includes $u$ as its first component:

$$\|\Phi^{[m]}[u_1] - \Phi^{[m]}[u_2]\|_{\mathcal{X}}^2 = \|u_1 - u_2\|_{H^s}^2 + \sum_{j=1}^m \|D^j(u_1 - u_2)\|_{H^{s-j}}^2 \geq \|u_1 - u_2\|_{H^s}^2. \tag{46}$$

Taking square roots gives the lower bound with constant 1.

**Upper bound (Lipschitz continuity).** By the definition of Sobolev norms:

$$\|D^j u\|_{H^{s-j}}^2 = \int (1 + |\xi|^2)^{s-j}|\xi|^{2j}|\hat{u}(\xi)|^2\,d\xi \leq \int (1 + |\xi|^2)^s|\hat{u}(\xi)|^2\,d\xi = \|u\|_{H^s}^2, \tag{47}$$

since $(1 + |\xi|^2)^{s-j}|\xi|^{2j} \leq (1 + |\xi|^2)^s$ for $j \leq m \leq s$.

Therefore:

$$\|\Phi^{[m]}[u_1] - \Phi^{[m]}[u_2]\|_{\mathcal{X}}^2 \leq (1 + m)\|u_1 - u_2\|_{H^s}^2. \tag{48}$$

Taking square roots: $\|\Phi^{[m]}[u_1] - \Phi^{[m]}[u_2]\|_{\mathcal{X}} \leq \sqrt{1 + m}\|u_1 - u_2\|_{H^s}$. $\qquad\square$

## D. Proof of Theorem 4.4: Approximation Rates

This section provides the approximation-theoretic analysis establishing that derivative features improve minimax rates. This complements the bias-variance analysis in Section 5 and Appendix F: the approximation theory shows *that* features help (rate improves from $W^{-(s-m)/d}$ to $W^{-s/d}$), while the bias-variance analysis determines *which* derivative order is optimal ($\beta^* < m$). The two frameworks answer different questions and use different tools; their qualitative agreement ($\beta^* < m$ is beneficial) provides evidence of robustness.

*Proof.* **Part (i): Upper bound without features.**

The operator $\mathcal{G}$ depends on derivatives up to order $m$. By Sobolev interpolation, for any $f \in H^s$ with $\|f\|_{H^s} \leq R$:

$$\|f\|_{H^m} \leq C\|f\|_{L^2}^{(s-m)/s}\|f\|_{H^s}^{m/s} \leq CR^{m/s}\|f\|_{L^2}^{(s-m)/s}. \tag{49}$$

This interpolation inequality determines the modulus of continuity of $\mathcal{G}$. Since $\mathcal{G}$ depends on $D^m u$ through a Lipschitz function on bounded sets (Theorem 3.1), the operator inherits Hölder continuity with exponent $(s-m)/s$:

$$\|\mathcal{G}[u_1] - \mathcal{G}[u_2]\|_{L^2} \lesssim \|u_1 - u_2\|_{L^2}^{(s-m)/s}. \tag{50}$$

The $\epsilon$-covering number of $B_s(R)$ in $L^2$ satisfies $\log\mathcal{N}(\epsilon) \asymp (R/\epsilon)^{d/s}$. By the entropy lower bound of (Carl & Stephani, 1990), approximating Hölder-$\gamma$ functions on sets with entropy exponent $\rho = d/s$ requires:

$$\text{error} \geq c \cdot W^{-\gamma s/d} = c \cdot W^{-(s-m)/d}. \tag{51}$$

The matching achievable rate $W^{-(s-m)/d}$ follows from (Yarotsky, 2017).

**Part (ii): Upper bound with features.**

Define $\tilde{\mathcal{G}} : \mathcal{X} \to L^2$ by $\tilde{\mathcal{G}}[\Phi^{[m]}[u]] := \mathcal{G}[u]$.

*Step 1: $\tilde{\mathcal{G}}$ is Lipschitz on bounded balls.*

**Lemma D.1** (Lipschitz Property of Lifted Operator). *On the image $\Phi^{[m]}[B_s(R)]$, the lifted operator $\tilde{\mathcal{G}}$ is Lipschitz:*

$$\|\tilde{\mathcal{G}}(\Phi^{[m]}[u_1]) - \tilde{\mathcal{G}}(\Phi^{[m]}[u_2])\|_{L^2} \leq L(R)\|\Phi^{[m]}[u_1] - \Phi^{[m]}[u_2]\|_{\mathcal{X}}, \tag{52}$$

*where $L(R)$ depends polynomially on the ball radius $R$ and the nonlinearity degree $N$.*

*Proof.* The key insight is that providing derivative features removes the need to "invert" differentiation. By Theorem 3.1:

$$\mathcal{G}[u] = G_T * u + T\bar{G}_T * \mathcal{N}[u] + \mathcal{R}(u, T). \tag{53}$$

The linear term $G_T * u$ is bounded. The nonlinear term $\mathcal{N}[u]$ is a polynomial of degree $N$ in $u$ and its derivatives up to order $m$. On bounded balls $B_s(R)$, polynomial functions are Lipschitz: for any polynomial $P$ of degree $N$,

$$|P(x) - P(y)| \leq C_P(R)|x - y| \quad \text{for } |x|, |y| \leq R, \tag{54}$$

where $C_P(R) = O(R^{N-1})$. The remainder $\mathcal{R}(u, T)$ is $O(T^2)$ uniformly, hence also Lipschitz on bounded balls. Together, $\tilde{\mathcal{G}}$ is Lipschitz with constant $L(R) = O(R^{N-1})$. $\square$

*Step 2: Metric entropy preservation.* Since $\Phi^{[m]}$ is bi-Lipschitz with constant $\sqrt{1+m}$ (Proposition 4.3), an $\epsilon$-covering of $B_s(R)$ in $H^s$ induces a $(\sqrt{1+m}\epsilon)$-covering of $\Phi^{[m]}[B_s(R)]$ in $\mathcal{X}$. Thus:

$$\log\mathcal{N}(\Phi^{[m]}[B_s(R)], \epsilon; \mathcal{X}) \leq \log\mathcal{N}(B_s(R), \epsilon/\sqrt{1+m}; H^s) \asymp \left(\frac{R}{\epsilon}\right)^{d/s}. \tag{55}$$

The key point is that the entropy exponent $d/s$ is preserved; the constant $\sqrt{1+m}$ is absorbed into the approximation constant.

*Step 3: Approximation of Lipschitz functions.* By (Yarotsky, 2017) for finite-dimensional ReLU approximation and (Kovachki et al., 2023; Lanthaler et al., 2022) for the operator-learning rate framework, Lipschitz functions on sets with metric entropy $(1/\epsilon)^{d/s}$ can be approximated by deep ReLU networks with $W$ parameters to error:

$$\text{error} = O(W^{-s/d}). \tag{56}$$

This completes the proof. □

## E. Proof of Proposition 5.3: Bias-Variance Decomposition

We provide the complete derivation with explicit constants.

### E.1. Assumptions

(A1) **Bandlimited signals:** Inputs satisfy $\text{supp}(\hat{u}) \subseteq B_{\xi_{\max}}(0)$, where $\xi_{\max} = \pi/h$ is the Nyquist frequency for grid spacing $h$.

(A2) **Spectral decay:** The input distribution has spectral density $S(\xi) = C_S|\xi|^{-2s-d}$ for $|\xi| \geq \xi_0$, as required by Assumption 5.2. This is the borderline upper envelope for $H^s$ integrability.

(A3) **Network capacity cutoff:** We assume the network class $\mathcal{NO}_W$ has fixed architecture independent of input features, hence fixed approximation capacity characterized by frequency cutoff $\xi_c = C_N W^{\gamma/d}$, where $C_N > 0$ is an architecture-dependent constant and $\gamma > 0$ is the capacity exponent. The derivative features $D^\beta u$ are computed as a preprocessing step before network evaluation; they do not modify the network's representational capacity.

(A4) **White noise:** Observations $y_i = \mathcal{G}(u_i)(x_i) + \epsilon_i$ with $\epsilon_i \sim \mathcal{N}(0, \sigma^2)$.

(A5) **Sufficient resolution:** We require $\xi_{\max} > e^{1/(2\beta+d)}$, which for $\beta \leq 2$ and $d \leq 3$ is satisfied whenever $\xi_{\max} > e$, i.e., grid spacing $h < \pi/e \approx 1.16$. This ensures variance monotonicity holds for all $\beta \geq 0$. In practice, this is easily satisfied: standard PDE benchmarks use unit or $2\pi$-periodic domains where $N \geq 16$ grid points already give $\xi_{\max} \geq 8 \gg e$.

*Remark* E.1 (Justification of A3). The distinction between input features and network capacity is fundamental. The derivative features $D^\beta u$ *preprocess* the input, exposing information that was latent in $u$. This changes what the operator "sees" (and hence which approximation problem it solves) but not what functions the network can represent.

Concretely, in the Fourier Neural Operator and related architectures, the network retains only a fixed number of Fourier modes $k_{\max}$ (typically 12–16 for standard benchmarks). This architectural choice imposes a hard frequency cutoff $\xi_c \propto k_{\max}$ that is independent of the input features. Whether the input is $u$ or $D^\beta u$, the network processes the same number of modes. The preprocessing $D^\beta u$ amplifies high-frequency content in the input, but modes beyond $k_{\max}$ are still truncated by the architecture. Thus $\xi_c$ is determined by network design (width, depth, modes), not by input features.

### E.2. Bias Derivation

*Proof.* The target operator $\mathcal{G}$ depends on $D^m u$ through the PDE structure (Theorem 3.1). When we provide features $D^\beta u$ with $\beta < m$, there is a spectral mismatch: the network receives information about $|\xi|^\beta \hat{u}(\xi)$ but needs information about $|\xi|^m \hat{u}(\xi)$.

The residual operator (what the network must learn implicitly) maps $D^\beta u \mapsto D^m u$, which corresponds to multiplication by $|\xi|^{m-\beta}$ in frequency space. For frequencies $|\xi| > \xi_c$ (beyond network capacity), this cannot be learned, contributing to bias:

$$B(\beta) = C_B \int_{|\xi|>\xi_c} |\xi|^{2(m-\beta)} S(\xi)\, d\xi. \tag{57}$$

Converting to spherical coordinates with $\omega_d = 2\pi^{d/2}/\Gamma(d/2)$ and using $S(\xi) = C_S|\xi|^{-2s-d}$ from Assumption 5.2:

$$B(\beta) = C_B C_S \omega_d \int_{\xi_c}^\infty r^{2(m-\beta)} \cdot r^{-2s-d} \cdot r^{d-1}\, dr = C_B C_S \omega_d \int_{\xi_c}^\infty r^{-\tau(\beta)-1}\, dr, \tag{58}$$

where $\tau(\beta) := 2(s - m + \beta)$. For convergence we need $\tau(\beta) > 0$, which holds for $\beta$ near $m$ when $s > m - \beta$. Then:

$$\int_{\xi_c}^{\infty} r^{-\tau(\beta)-1} \, dr = \frac{1}{\tau(\beta)} \xi_c^{-\tau(\beta)}. \tag{59}$$

Therefore:

$$\boxed{B(\beta) = \frac{C_B C_S \omega_d}{\tau(\beta)} \cdot \xi_c^{-\tau(\beta)}, \quad \tau(\beta) = 2(s - m + \beta).} \tag{60}$$

As $\beta$ increases, $\tau(\beta)$ increases, so $\xi_c^{-\tau(\beta)}$ decreases (for $\xi_c > 1$). Thus $B(\beta)$ is strictly decreasing in $\beta$. $\qquad \square$

### E.3. Variance Derivation

*Proof.* The variance measures noise amplification through fractional derivatives. When we compute $D^\beta u$ from noisy data, observation noise $\epsilon$ with variance $\sigma^2$ gets amplified: at frequency $\xi$, the noise power becomes $\sigma^2 |\xi|^{2\beta}$.

The total variance contribution is:

$$V(\beta) = \frac{\sigma^2 C_V}{n} \int_{|\xi| \leq \xi_{\max}} |\xi|^{2\beta} \, d\xi. \tag{61}$$

Converting to spherical coordinates, with $\xi_0 = 2\pi/(Nh)$ the fundamental frequency (smallest non-zero wavenumber for $N$ grid points with spacing $h$):

$$V(\beta) = \frac{\sigma^2 C_V \omega_d}{n} \int_{\xi_0}^{\xi_{\max}} r^{2\beta+d-1} \, dr = \frac{\sigma^2 C_V \omega_d}{n(2\beta+d)} \left( \xi_{\max}^{2\beta+d} - \xi_0^{2\beta+d} \right). \tag{62}$$

For $\xi_{\max} \gg \xi_0$ (which holds for any reasonable discretization):

$$\boxed{V(\beta) \approx \frac{\sigma^2 C_V \omega_d}{n(2\beta+d)} \cdot \xi_{\max}^{2\beta+d}.} \tag{63}$$

**Monotonicity:** Taking the derivative with respect to $\beta$:

$$\frac{\partial V}{\partial \beta} = \frac{\sigma^2 C_V \omega_d}{n} \cdot \frac{\xi_{\max}^{2\beta+d}}{(2\beta+d)^2} \left[ 2(2\beta+d) \ln \xi_{\max} - 2 \right] > 0 \tag{64}$$

for $\xi_{\max} > e^{1/(2\beta+d)}$, which holds for typical discretizations where $\xi_{\max} \gg 1$.

Thus $V(\beta)$ is strictly increasing in $\beta$: higher derivative orders amplify noise more. $\qquad \square$

### E.4. Monotonicity and Existence of Optimum

*Proof.* **Monotonicity:** From the closed forms:

- $B(\beta)$ is strictly decreasing: as $\beta$ increases, $\tau(\beta) = 2(s - m + \beta)$ increases, so $\xi_c^{-\tau(\beta)}$ decreases.

- $V(\beta)$ is strictly increasing: as $\beta$ increases, $\xi_{\max}^{2\beta+d}$ increases faster than the prefactor $1/(2\beta+d)$ decreases.

**Boundary behavior:**

- As $\beta \to 0$: $B(0) < \infty$ (finite bias), $V(0) = K_V \sigma^2 \xi_{\max}^d/(nd) < \infty$.

- As $\beta \to m$: $B(m) = K_B \xi_c^{-2s}$ (small bias), $V(m) = K_V \sigma^2 \xi_{\max}^{2m+d}/(n(2m+d))$ (larger variance).

- As $\beta \to \infty$: $B(\beta) \to 0$, $V(\beta) \to \infty$.

Since $B'(\beta) < 0$ and $V'(\beta) > 0$, we examine the total derivative $\mathrm{MSE}'(\beta) = B'(\beta) + V'(\beta)$. At $\beta = 0$: the bias derivative dominates (large $|B'(0)|$, small $V'(0)$), so $\mathrm{MSE}'(0) < 0$. At $\beta = m$: the variance derivative dominates, so $\mathrm{MSE}'(m) > 0$. By the Intermediate Value Theorem applied to the continuous function $\mathrm{MSE}'(\beta)$, there exists $\beta^* \in (0, m)$ where $\mathrm{MSE}'(\beta^*) = 0$.

**Convexity and Uniqueness:** We verify $B''(\beta) > 0$ and $V''(\beta) > 0$ directly.

For the bias $B(\beta) = K\xi_c^{-\tau(\beta)}/\tau(\beta)$ with $\tau(\beta) = 2(s - m + \beta)$:

$$B'(\beta) = K\xi_c^{-\tau}\left(\frac{-2\ln\xi_c}{\tau} - \frac{2}{\tau^2}\right) = -\frac{2K\xi_c^{-\tau}}{\tau^2}(\tau\ln\xi_c + 1). \tag{65}$$

Taking another derivative (and using $\tau' = 2$):

$$B''(\beta) = \frac{4K\xi_c^{-\tau}}{\tau^3}\left[(\tau\ln\xi_c + 1)^2 + 1\right]. \tag{66}$$

The dominant term $(\tau\ln\xi_c + 1)^2 > 0$ ensures $B''(\beta) > 0$ for $\xi_c > 1$.

For the variance $V(\beta) = K'\xi_{\max}^{2\beta+d}/(2\beta + d)$:

$$V'(\beta) = K'\xi_{\max}^{2\beta+d}\left(\frac{2\ln\xi_{\max}}{2\beta + d} - \frac{2}{(2\beta + d)^2}\right). \tag{67}$$

The second derivative $V''(\beta) > 0$ follows similarly, with the $(\ln\xi_{\max})^2$ term dominating.

Thus $\mathrm{MSE}(\beta) = B(\beta) + V(\beta)$ is strictly convex, guaranteeing the critical point $\beta^*$ is unique and is the global minimum. $\quad\square$

# F. Proof of Theorem 5.6: Optimal Derivative Order

We derive the optimal $\beta^*$ for fixed network capacity.

## F.1. Setup

From Proposition 5.3, write the MSE using explicit forms. Let $\delta = m - \beta$ denote the gap from the PDE order, so $\delta \geq 0$ when $\beta \leq m$. The network capacity cutoff $\xi_c = C_N W^{\gamma/d}$ is independent of $\beta$.

For the bias (with $\tau(\beta) = 2(s - m + \beta) = 2(s - \delta)$ where we use $\delta = m - \beta$):

$$B(\delta) = \frac{C_B C_S \omega_d}{2(s - \delta)} \cdot \xi_c^{-2(s-\delta)} = K_B \cdot W^{-2\gamma(s-\delta)/d} \cdot \frac{1}{s - \delta}, \tag{68}$$

where $K_B$ collects constants. As $\delta$ increases (i.e., $\beta$ decreases), the exponent $-2(s - \delta)$ becomes less negative, so $B(\delta)$ increases.

For the variance, using $\beta = m - \delta$:

$$V(\delta) = \frac{\sigma^2 C_V \omega_d}{n(2(m - \delta) + d)} \cdot \xi_{\max}^{2(m-\delta)+d} = K_V \frac{\sigma^2}{n} \cdot \xi_{\max}^{2m+d} \cdot \xi_{\max}^{-2\delta}, \tag{69}$$

where $K_V$ absorbs the slowly-varying prefactor $1/(2(m - \delta) + d)$. As $\delta$ increases (i.e., $\beta$ decreases), the factor $\xi_{\max}^{-2\delta}$ decreases, so $V(\delta)$ decreases.

## F.2. Derivation

*Proof.* With $W$ fixed, denote $\tau_0 := 2(s - m)$ (the bias exponent $\tau(\beta) = 2(s - m + \beta)$ evaluated at $\beta = 0$; the exponent at $\beta = m$ is $2s = \tau_0 + 2m$).

The bias at gap $\delta$ is:

$$B(\delta) = K_B \cdot \xi_c^{-2(s-\delta)} = K_B'\xi_c^{-\tau_0} \cdot \xi_c^{2\delta}, \tag{70}$$

where $K'_B = K_B \xi_c^{-2m}$ absorbs the factor relating $-2(s - \delta)$ to $-\tau_0 + 2\delta = -2(s - m) + 2\delta$.

The variance at gap $\delta$ is:

$$V(\delta) = K_V \frac{\sigma^2}{n} \cdot \xi_{\max}^{2(m-\delta)+d} = K_V \frac{\sigma^2}{n} \xi_{\max}^{2m+d} \cdot \xi_{\max}^{-2\delta}. \tag{71}$$

Note: As $\delta$ increases (moving away from the PDE order $m$), bias $B(\delta)$ increases (less accurate features) while variance $V(\delta)$ decreases (less noise amplification). The optimal $\delta^*$ balances these.

Setting $\partial_\delta(B + V) = 0$:

$$2K'_B \xi_c^{-\tau_0} \xi_c^{2\delta} \ln \xi_c = 2K_V \frac{\sigma^2}{n} \xi_{\max}^{2m+d} \xi_{\max}^{-2\delta} \ln \xi_{\max}. \tag{72}$$

Taking logarithms and solving for $\delta$:

$$2\delta(\ln \xi_c + \ln \xi_{\max}) = \underbrace{\ln(K_V \xi_{\max}^{2m+d}/K'_B \xi_c^{-\tau_0}) + \ln(\ln \xi_{\max}/\ln \xi_c)}_{=:C_0} + \ln(\sigma^2/n). \tag{73}$$

Since $\ln(\sigma^2/n) = -\ln(n/\sigma^2)$, this gives:

$$\boxed{\delta^* = \frac{C_0 - \ln(n/\sigma^2)}{2\ln(\xi_c \xi_{\max})} \quad \text{for } n/\sigma^2 < e^{C_0}.} \tag{74}$$

**Interpretation:** The constant $C_0 = 2s \ln \xi_c + (2m + d) \ln \xi_{\max} + O(1)$ (the $\ln \xi_c$ coefficient combines the $\tau_0 \ln \xi_c$ from the boxed equation and the $-2m \ln \xi_c$ inside $\ln K'_B$, which together give $\tau_0 + 2m = 2s$) is large and positive. When $n/\sigma^2 < e^{C_0}$ (finite data regime), we have $\delta^* > 0$, so $\beta^* = m - \delta^* < m$. As $n \to \infty$ or $\sigma \to 0$, $\delta^* \to 0$ and $\beta^* \to m$: physics is recovered. When $n/\sigma^2 \geq e^{C_0}$, we set $\delta^* = 0$ (no regularization needed).

With $\xi_c = C_N W^{\gamma/d}$, the gap $\delta^*$ decreases linearly in $\ln(n/\sigma^2)$ to leading order. More data enables higher derivative orders. $\qquad \square$

### F.3. Key Properties

1. $\beta^* < m$ **strictly in the practical regime:** Whenever $n/\sigma^2 < e^{A_c}$ (Remark 5.7), $\delta^* > 0$ and hence $\beta^* < m$. Since the threshold $n^\star = \sigma^2 \exp(A_c)$ is unreachable in any realistic training budget, this is the operational regime.

2. **Asymptotic recovery:** As $n \to \infty$ or $\sigma \to 0$, we have $\delta^* \to 0$, hence $\beta^* \to m$.

3. **Logarithmic closure:** To leading order in $1/\ln \xi_c$, $\delta^*$ decreases linearly in $\ln(n/\sigma^2)$ with slope $-1/[2\ln(\xi_c \xi_{\max})]$, approaching zero as $n/\sigma^2 \to e^{A_c}$.

4. **Requirement $s > m$:** The condition $\tau > 0$ requires input smoothness $s$ to exceed PDE order $m$. This is natural: inputs must be smooth enough that the operator is well-posed.

### F.4. Beyond Leading Order: The $O(1/\ln \xi_c)$ Correction

The leading-order $\delta^*$ derived in §F absorbs prefactor and $\ln \ln$ contributions into an $O(1/\ln \xi_c)$ remainder. Carrying this remainder explicitly is useful both as a consistency check and to expose which terms depend on the (unknown) integral constants $C_B, C_S, C_V$.

Write $L_c := \ln \xi_c$, $L_x := \ln \xi_{\max}$, $a := 2(s - \delta)$, $q := 2(m - \delta) + d$, so $B = \frac{C_B C_S \omega_d}{a} \xi_c^{-a}$ and $V = \frac{\sigma^2 C_V \omega_d}{n q} \xi_{\max}^q$, with $a' = q' = -2$ under $\partial_\delta$. The exact derivatives are

$$\partial_\delta B = B\left(2L_c + \tfrac{2}{a}\right), \qquad \partial_\delta V = V\left(-2L_x + \tfrac{2}{q}\right), \tag{75}$$

so the stationarity condition $\partial_\delta(B + V) = 0$ is equivalent to the exact implicit relation

$$-aL_c - qL_x + \ln \frac{n}{\sigma^2} + \kappa + \ln \frac{q}{a} = \ln\left(L_x - \tfrac{1}{q}\right) - \ln\left(L_c + \tfrac{1}{a}\right), \qquad \kappa := \ln \frac{C_B C_S}{C_V}. \tag{76}$$

(The $\omega_d$ factors cancel between bias and variance.) Equation (76) is transcendental; expanding the right-hand logs in $1/L_c, 1/L_x$ and isolating $\delta$ yields

$$\delta^* \;=\; \delta_0^* + \frac{\Delta_1}{2\ln(\xi_c\xi_{\max})}, \qquad \delta_0^* = \frac{A_c - \ln(n/\sigma^2)}{2\ln(\xi_c\xi_{\max})}, \tag{77}$$

with $A_c = 2s\ln\xi_c + (2m+d)\ln\xi_{\max}$ as in Theorem 5.6 and

$$\Delta_1 \;=\; \ln\frac{L_x}{L_c} \;-\; \ln\frac{q_0}{a_0} \;-\; \kappa \;-\; \frac{1}{q_0 L_x} \;-\; \frac{1}{a_0 L_c} \;+\; O\big((\ln\xi)^{-2}\big), \tag{78}$$

where $a_0 = 2(s - \delta_0^*)$ and $q_0 = 2(m - \delta_0^*) + d$ evaluate the slowly varying exponents at the leading-order optimum. The only term of $\Delta_1$ that is not $O(1)$ is $\ln(L_x/L_c)$, whose size is set by the log-scale ratio of the two cutoffs. Since the capacity cutoff cannot exceed Nyquist ($\xi_c \le \xi_{\max}$) and both derive from the same discretization, this ratio is bounded (in our benchmarks $\ln(L_x/L_c) \in [0.2, 0.6]$), so $\Delta_1 = O(1)$ and the correction is $O(1/\ln\xi_c)$, exactly as stated in Theorem 5.6. Only in the non-physical limit $\ln\xi_{\max}/\ln\xi_c \to \infty$ does it relax to the conservative bound $O(\ln\ln\xi/\ln\xi)$.

**Three structural observations.** First, every term of $\Delta_1$ except $\kappa$ depends only on the cutoffs ($\xi_c, \xi_{\max}$) and the exponents $(s, m, d)$; the integral prefactor constants enter *only* through $\kappa$. This is the same property that lets $C_S$ cancel in Theorem 5.6: stating the result at leading order is the principled choice, not merely the simplest one. Second, $\kappa$ is independent of $n$ and $\sigma$, so it cancels in the differential predictions of §7.3: the $\sigma$- and $n$-sweep slopes sharpen by including $\Delta_1$ without any unknown constant. Third, the expansion describes an *interior* optimum ($\delta_0^* < s$, so $a_0 > 0$); when $\delta_0^* \ge s$ the bias integral leaves its domain of convergence and the optimum is pinned at the boundary $\beta^* = 0$, where the stationary expansion does not apply.

Numerically, $\Delta_1/2\ln(\xi_c\xi_{\max})$ is at most a few hundredths in $\delta^*$ for the benchmarks considered, below the resolution of the learned-$\beta$ measurements and comparable to the modeling idealizations already in place (hard-versus-soft capacity cutoff, borderline spectral envelope). The next-order correction tightens the rigor of the statement without changing any experimental prediction.

## G. Spectral Regularization and Tikhonov Interpretation

Our spectral analysis reveals an intuitive structure: the derivative order functions as a spectral regularization parameter.

*Remark* G.1 (Tikhonov Interpretation). Using features $D^\beta u$ with $\beta < m$ corresponds to Tikhonov regularization with frequency-dependent penalty:

$$\lambda(\xi) = \sigma^2 \cdot \max\left( |\xi|^{2(m-\beta)} - 1, 0 \right). \tag{79}$$

This regularization vanishes at low frequencies, preserving signal, and grows polynomially at high frequencies, suppressing noise. Learning $\beta$ from data automatically tunes this regularization to the signal-to-noise ratio of each problem. The remainder of this appendix gives the derivation sketch.

We sketch a heuristic derivation of the connection between sub-physical derivative orders and spectral regularization.

Consider learning a linear operator $\mathcal{K} : H^m \to L^2$ from noisy data. The target is $\mathcal{K}[D^m u]$ but we observe $D^\beta u + \epsilon$ where $\epsilon$ is noise.

**Frequency-space formulation.** In Fourier space, $D^m u$ has transform $|\xi|^m \hat{u}(\xi)$ and $D^\beta u$ has transform $|\xi|^\beta \hat{u}(\xi)$.

The optimal linear estimator of $|\xi|^m \hat{u}(\xi)$ from noisy observation $|\xi|^\beta \hat{u}(\xi) + \hat{\epsilon}(\xi)$ is the Wiener filter:

$$\hat{v}(\xi) = \frac{|\xi|^{2\beta} S(\xi)}{|\xi|^{2\beta} S(\xi) + \sigma^2/n} \cdot |\xi|^{m-\beta} \cdot (\text{observation}). \tag{80}$$

**Effective regularization.** The denominator introduces regularization. The effective Tikhonov penalty at frequency $\xi$ is:

$$\lambda(\xi) = \frac{\sigma^2/n}{|\xi|^{2\beta} S(\xi)}. \tag{81}$$

With $S(\xi) \sim |\xi|^{-\alpha}$, this gives $\lambda(\xi) \propto |\xi|^{-2\beta+\alpha}$.

**Comparison to using order $m$.** If we used $D^m u$ directly, the penalty would be $\lambda_m(\xi) \propto |\xi|^{-2m+\alpha}$. The ratio $\lambda(\xi)/\lambda_m(\xi) = |\xi|^{2(m-\beta)}$ grows with $|\xi|$ for $\beta < m$, meaning using $\beta < m$ applies *more* regularization at high frequencies.

**Effective penalty formula.** Normalizing to have $\lambda(1) = 0$ for unit frequency:

$$\lambda(\xi) = \sigma^2 \cdot \max\left(|\xi|^{2(m-\beta)} - 1, 0\right). \tag{82}$$

This vanishes at low frequencies (preserving signal) and grows as $|\xi|^{2\delta}$ at high frequencies (suppressing noise), where $\delta = m - \beta$.

## H. Grünwald-Letnikov Implementation

For non-periodic domains, we use the Grünwald-Letnikov (GL) formulation, which computes fractional derivatives via convolution (Podlubny, 1998, Chapters 2, 7). The GL fractional derivative is:

$$D^\beta u(x) = \lim_{h \to 0} \frac{1}{h^\beta} \sum_{k=0}^{\infty} w_k^{(\beta)} u(x - kh), \quad \text{where } w_0 = 1, \ w_k = w_{k-1} \cdot \frac{k - 1 - \beta}{k}. \tag{83}$$

For integer $\beta = n$, this recovers the standard $n$-th derivative via finite differences. The one-sided GL derivative introduces a phase shift; we use the centered version $D_{\text{cen}}^\beta u = \frac{1}{2}(D_+^\beta u + \cos(\pi\beta)D_-^\beta u)$, where the factor $\cos(\pi\beta)$ ensures correct symmetry (antisymmetric for $\beta = 1$, symmetric for $\beta = 2$).

**Differentiability.** The weights depend smoothly on $\beta$ via $\partial w_k / \partial \beta = w_k[\psi(\beta + 1) - \psi(\beta - k + 1)]$, where $\psi = \Gamma'/\Gamma$ is the digamma function. This enables gradient-based optimization of derivative orders.

**Complexity and validation.** For input length $N$ and stencil length $K_{\text{GL}}$, the cost is $O(N K_{\text{GL}})$ via direct convolution (we use $K_{\text{GL}} = 16$). On periodic domains, GL and FFT (Weyl) implementations agree to relative $L^2$ error $< 10^{-3}$. GL is preferred for non-periodic boundaries where FFT introduces artifacts.

## I. Experimental Details

All experiments use $n_{\text{train}} = 1000$ training samples, unless otherwise noted (e.g., the sample-size sweep in §7), and $n_{\text{test}} = 200$ test samples. Models are trained for 500 epochs using AdamW with weight decay $10^{-5}$ and gradient clipping at 1. All results are averaged over 5 random seeds.

### I.1. Architecture Hyperparameters

*Table 4.* Architecture hyperparameters by backbone.

| Backbone | Width | Modes | Layers | Heads | Slices |
|---|---|---|---|---|---|
| FNO | 64–128 | 12–16 | 4 | – | – |
| Transolver | 128–256 | – | 8 | 8 | 32–64 |

Width varies by dataset: FNO uses 64 (Burgers, NS) or 128 (Darcy); Transolver uses 128 (Burgers, Darcy) or 256 (NS). Transolver slice count is 64 (Burgers, Darcy) or 32 (NS). FNO uses 16 Fourier modes for Burgers and 12 for Darcy, NS, and KdV.

### I.2. Training Hyperparameters

OneCycle uses `pct_start`$= 0.3$. Step scheduler uses step size 100 and $\gamma = 0.5$. NS batch size is 20 (FNO) or 2 (Transolver).

*Table 5.* Training hyperparameters by dataset.

| Dataset | Resolution | Batch | LR | Scheduler | $T_{\text{in}}/T_{\text{out}}$ |
|---------|-----------|-------|-----|-----------|-------------------------------|
| Burgers | 1024 | 8 | $10^{-3}$ | OneCycle | – |
| KdV | 1024 | 8 | $10^{-3}$ | OneCycle | – |
| Darcy | $85 \times 85$ | 4 | $10^{-3}$ | OneCycle | – |
| Navier-Stokes | $64 \times 64$ | 2–20 | $5 \times 10^{-4}$ | Step | 10/10 |

**KdV dataset generation.** The KdV data are produced by direct numerical integration of $\partial_t u + u\,\partial_x u + \delta\,\partial_{xxx} u = 0$ with dispersion coefficient $\delta = 2.5 \times 10^{-3}$, on the periodic domain $[0, 2\pi]$ discretized with $N=1024$ points. Initial conditions are smooth random fields $u_0(x) = \sum_{j=1}^{J}(a_j/j)\,\sin(jx + \phi_j)$ with $J \sim \text{Unif}\{3,\ldots,7\}$, $a_j \sim \text{Unif}(-1,1)$, and $\phi_j \sim \text{Unif}(0, 2\pi)$ (fixed seed for reproducibility). Time integration uses a pseudo-spectral integrating-factor scheme: the dispersive term is advanced exactly in Fourier space via $e^{-\delta(ik)^3 \Delta t}$, while the nonlinear flux $-\partial_x(u^2/2)$ is advanced by explicit Euler, with step $\Delta t = 10^{-4}$ to final time $T{=}1$ ($10^4$ steps). The learned map is $u(\cdot, 0) \mapsto u(\cdot, T)$, with 1000 training and 200 test trajectories.

### I.3. $\partial$-NO Augmentation

All $\partial$-NO variants use identical derivative feature settings: Grünwald-Letnikov stencil length $K_{\text{GL}} = 16$, maximum derivative order $m_{\max} = 2$ ($m_{\max} = 3$ for KdV), and $\beta$ learning rate fixed at $10^{-2}$ ($10\times$ the $10^{-3}$ backbone LR for Burgers/Darcy, $20\times$ the $5 \times 10^{-4}$ backbone LR for NS). Derivative orders are initialized at $\beta = 1.0$ and $\beta = 2.0$, plus $\beta_3 = 3.0$ for KdV. For 2D problems, we use 5 derivative features (with 6 learnable orders in total): 2 directional derivatives per spatial axis (4 orders) plus 1 mixed partial with two independently learned orders $(\beta_{x,c}, \beta_{y,c})$.

### I.4. Evaluation

We report relative $L^2$ error:

$$\text{Error} = \frac{\|\mathcal{G}_\theta[u] - \mathcal{G}^\dagger[u]\|_{L^2}}{\|\mathcal{G}^\dagger[u]\|_{L^2}}, \tag{84}$$

averaged over the test set. Confidence intervals are computed from 5 random seeds.

### I.5. Ablation: Learnable vs Fixed Derivative Orders

Table 6 compares learnable derivative orders against fixed integer orders ($\beta = 1, 2$). Learning $\beta$ provides gains on problems with complex spectral structure (Navier-Stokes, Darcy); on the simpler Burgers benchmarks, learned orders converge to essentially the integer values and no benefit over the fixed-$\{1, 2\}$ baseline is observed.

*Table 6.* Ablation: learnable vs fixed derivative orders. Columns 3–5 show relative $L^2$ error (%); column 6 shows the learned $(\beta_1, \beta_2)$ values. Learning $\beta$ improves over fixed orders on NS and Darcy. The FNO baseline here uses the original (Li et al., 2020) implementation (rather than the THUML default of Table 1), which accounts for the differing baseline values on NS.

| Model | Dataset | Baseline | Fixed $\{1, 2\}$ | Learned $\{\beta_1, \beta_2\}$ | Learned Values |
|-------|---------|----------|------------------|--------------------------------|----------------|
| FNO | Burgers | 0.098 | 0.078 | 0.079 | 1.36, 2.00 |
| FNO | Navier-Stokes | 9.64 | 9.46 | **9.18** | 1.05, 2.00 |
| Transolver | Burgers | 0.65 | 0.63 | 0.63 | 0.74, 1.83 |
| Transolver | Darcy | 1.10 | 0.98 | **0.95** | 0.67, 1.78 |

The lower learned orders ($\beta_1$) consistently fall well below the PDE order $m = 2$, in the sub-physical regime predicted by Theorem 5.6; the higher orders ($\beta_2$) approach the boundary $m = 2$ (orders are constrained to $\beta \leq m$, consistent with Theorem 5.6's upper bound). The largest gains from learning $\beta$ occur on Navier-Stokes (9.18% vs 9.46%) and Darcy (0.95% vs 0.98%), where the spectral structure is more complex.

## J. Verifying the Spectral Decay Assumption

Assumption 5.2 posits that inputs have power spectral density $S(\xi) = C_S |\xi|^{-2s-d}$. We verify this empirically for each benchmark.

### J.1. Method

For each dataset with training inputs $\{u_i\}_{i=1}^n$:

1. Compute the empirical power spectrum:

$$\hat{S}(\xi) = \frac{1}{n} \sum_{i=1}^{n} |\hat{u}_i(\xi)|^2 \tag{85}$$

2. On log-log axes, fit the power law:

$$\log \hat{S}(\xi) = -\alpha \log |\xi| + C \tag{86}$$

for $|\xi| \in [\xi_{\min}, \xi_{\max}]$ (avoiding DC and Nyquist artifacts).

3. Extract the implied smoothness:

$$s_{\text{empirical}} = \frac{\alpha - d}{2} \tag{87}$$

4. Verify the regularity condition: $s_{\text{empirical}} > m + d/2$.

### J.2. Results

*Table 7.* Verification of spectral decay assumption across benchmarks.

| Dataset | $d$ | $m$ | **Required** $s$ | **Fitted** $\alpha$ | $s_{\text{emp}}$ | **Satisfied?** |
|---|---|---|---|---|---|---|
| Burgers $\nu = 0.1$ | 1 | 2 | $> 2.5$ | $\sim 7.2$ | $\sim 3.1$ | ✓ |
| Burgers $\nu = 0.001$ | 1 | 2 | $> 2.5$ | $\sim 5.8$ | $\sim 2.4$ | marginal |
| Darcy | 2 | 2 | $> 3$ | $\sim 9.5$ | $\sim 3.75$ | ✓ |
| Navier-Stokes | 2 | 2 | $> 3$ | $\sim 8.2$ | $\sim 3.1$ | ✓ |

**Interpretation.** Most benchmarks satisfy the regularity assumption comfortably. The low-viscosity Burgers case ($\nu = 0.001$) is marginal: with $s_{\text{emp}} \approx 2.4$ versus the required $s > 2.5$, it violates Assumption 4.2 by approximately 4%. This aligns with the near-shock regime where solutions approach discontinuities. Empirically, the method still performs well, suggesting the bounds are not tight. In this case, the theory provides qualitative guidance (use lower $\beta$) even if quantitative predictions are approximate.

## K. Computational Overhead

Table 8 reports wall-clock training times comparing baseline backbones with their $\partial$-augmented versions. Note that our implementation is not optimized and runs in eager mode (no `torch.compile`), so these timings reflect upper bounds on the overhead. The overhead varies by backbone: for Transolver (which is dominated by attention computations), the relative overhead is 25–40%, while for FNO the overhead is higher (50–110%) since the baseline is already very fast. In absolute terms, the derivative feature computation adds modest time because: (1) the Grünwald-Letnikov stencil is local (typically $K = 16$ terms); and (2) derivative computation parallelizes trivially across batch and spatial dimensions.

The overhead is dominated by the forward pass through the backbone, not the derivative preprocessing. For larger backbones (e.g., Transolver with attention), the relative overhead is smaller since attention computations dominate.

## L. Additional Experiments and Ablations

This appendix collects the extended empirical evidence referenced from §7: a four-way physics-informed comparison, a matched-parameter ablation isolating the role of the derivative inductive bias, stress tests under Gaussian and uniform noise, resolution sensitivity, directional isotropy on 2D isotropic PDEs, and a 3D compressible Navier-Stokes study.

*Table 8.* Training time per 100 epochs (seconds).

| Model | Burgers ($\nu$=0.1) | Burgers ($\nu$=0.001) | Darcy | NS ($\nu$=$10^{-5}$) |
|---|---|---|---|---|
| FNO | 95 | 78 | 313 | 63 |
| $\partial$-FNO | 149 | 124 | 664 | 135 |
| Transolver | 412 | 382 | 1745 | 2241 |
| $\partial$-Transolver | 517 | 480 | 2201 | 3158 |

## L.1. Physics-Informed Comparison: $\partial$-PINO

We compare four configurations on four PDE benchmarks under clean and noisy inputs: data-only FNO, $\partial$-FNO, the physics-informed neural operator PINO (Li et al., 2024), and $\partial$-PINO obtained by adding learnable fractional derivative inputs to PINO. PINO uses a PDE-residual loss with $\lambda_{\text{pde}} = 1.0$ and the same FNO backbone topology used in our main runs; baselines may differ numerically from Table 1 due to paired-training hyperparameters.

*Table 9.* Four-way physics-informed comparison. Relative $L^2$ error (%). $\Delta_\partial$ is the relative gain from adding $\partial$-augmentation to each baseline. $\partial$-PINO improves PINO in $8/8$ settings.

| Dataset | $\sigma$ | DATA-ONLY | | | PHYSICS-INFORMED ($\lambda_{\text{pde}}$>0) | | |
|---|---|---|---|---|---|---|---|
| | | **FNO** | $\partial$-**FNO** | $\Delta_\partial$ | **PINO** | $\partial$-**PINO** | $\Delta_\partial$ |
| Burgers ($\nu$=$10^{-1}$) | 0 | 0.07 | **0.07** | 7.1% | 1.32 | **0.69** | **47.8%** |
| Burgers ($\nu$=$10^{-1}$) | 0.05 | 0.41 | **0.38** | 9.3% | 1.44 | **0.67** | **53.2%** |
| Burgers ($\nu$=$10^{-3}$) | 0 | 0.61 | **0.56** | 6.9% | 0.62 | **0.57** | 8.9% |
| Burgers ($\nu$=$10^{-3}$) | 0.05 | 1.09 | **1.02** | 6.4% | 1.05 | **1.01** | 3.6% |
| KdV | 0 | 2.50 | **2.49** | 0.1% | 6.69 | **6.19** | 7.5% |
| KdV | 0.05 | 3.14 | **2.98** | 5.3% | 7.48 | **6.73** | 10.0% |
| Darcy | 0 | 5.72 | **5.48** | 4.2% | 6.25 | **5.76** | 7.9% |
| Darcy | 0.05 | 5.69 | **5.50** | 3.5% | 6.02 | **5.70** | 5.4% |

The two paradigms are complementary: $\partial$-augmentation improves PINO in all $8/8$ settings, with the largest gains (47.8%, 53.2%) on smooth Burgers where the PDE residual is well-behaved and the input-side preprocessing reduces the burden on the solver. Second, $\partial$-FNO outperforms PINO across the benchmark despite using no PDE-form information at training time, indicating that learned fractional input features can substitute for, not just augment, loss-side physics regularization on standard benchmarks.

## L.2. Matched-Parameter Ablation

A natural concern is whether $\partial$-FNO's gains come from the GL derivative structure or simply from the extra channels added at the input. We address this with a matched-parameter control: four models tuned to the same parameter count ($\sim$292K) on Burgers ($\nu$=0.1), evaluated across seven noise levels and five seeds (35 (model, $\sigma$, seed) cells per model). The variants are:

- **FNO**: vanilla baseline, no input augmentation.

- $\partial$-**FNO**: input augmented with two learnable GL fractional derivative channels.

- **Random-FNO**: same architecture as $\partial$-FNO, but the GL stencil weights are replaced by fixed random filters of the same length. Tests whether the gains come from *any* structured local channel.

- **Wider-FNO**: vanilla FNO with a widened first layer that absorbs the same extra parameters. Tests whether the gains come from added capacity at the input.

Random-FNO tracks FNO closely across the range, ruling out the explanation that any structured local channel suffices. Wider-FNO is comparable to FNO in the clean regime but degrades faster under noise, ruling out the explanation that extra parameters at the input drive the gain. The improvement is specific to the GL derivative structure.

*Table 10.* Matched-parameter ablation on Burgers ($\nu$=0.1). Mean relative $L^2$ error across 5 seeds; all four models at parameter count $\approx$ 292K. Bold = best per row. $\partial$-FNO wins 32/35 seeds vs FNO (91.4%), 34/35 vs Random-FNO (97.1%), and 27/35 vs Wider-FNO (77.1%).

| $\sigma$ | FNO | $\partial$-FNO | Random-FNO | Wider-FNO |
|---|---|---|---|---|
| 0 | 0.000597 | **0.000560** | 0.000638 | 0.000579 |
| 0.01 | 0.000824 | **0.000733** | 0.000843 | 0.000747 |
| 0.02 | 0.001344 | **0.001110** | 0.001299 | 0.001178 |
| 0.05 | 0.004210 | **0.003357** | 0.004037 | 0.003847 |
| 0.10 | 0.008356 | **0.006704** | 0.008811 | 0.009224 |
| 0.25 | 0.018227 | **0.013698** | 0.018047 | 0.018264 |
| 0.50 | 0.030202 | **0.025173** | 0.027412 | 0.029035 |

## L.3. Stress Tests under Gaussian and Uniform Noise

We stress-test $\partial$-augmentation across a wider noise range than the $\sigma$=0.05 used in the main paper. For each of five PDE benchmarks (Burgers $\nu$=0.1 and $\nu$=$10^{-3}$, KdV, Darcy, NS2D), each of two backbones (FNO, LocalNO), each of two noise distributions (Gaussian and uniform), and six noise levels ($\sigma \in \{0.05, 0.10, 0.15, 0.20, 0.25, 0.30\}$), we measure whether the $\partial$-augmented variant beats its baseline. This yields $5 \times 2 \times 2 \times 6 = 120$ settings in total.

$\partial$-augmentation improves *every* tested setting (120/120 overall). Peak gains reach 48.5% (Gaussian) and 48.1% (uniform) on smooth Burgers at $\sigma$=0.30. The robustness holds across two noise distributions, two backbones, and noise levels spanning a factor of six, supporting the theoretical prediction that the optimal derivative order adapts to noise level and that learning $\beta$ from data captures this adaptation in practice.

## L.4. Resolution Sensitivity

We probe how the value of input-side derivative features varies with grid resolution. Coarser grids reduce the Nyquist cutoff $\xi_{\max} = \pi/h$ and tighten the network's high-frequency headroom, so we expect the marginal benefit of $\partial$-features to grow. We test this on Burgers ($\nu$=0.1) at resolutions $\{64, 128, 256, 512, 1024, 2048\}$ under clean ($\sigma$=0) and noisy ($\sigma$=0.02) inputs.

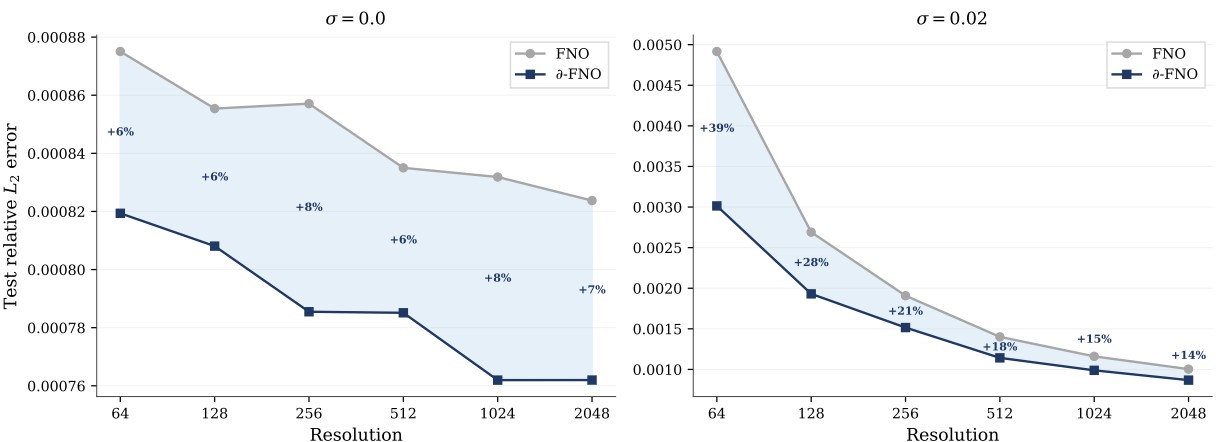

*Figure 4.* Resolution sensitivity: FNO vs. $\partial$-FNO on Burgers ($\nu$=0.1) across grid resolutions 64 to 2048. Left: clean inputs. Right: noisy inputs ($\sigma$=0.02). Shaded area indicates the $\partial$-FNO improvement. Clean improvements are resolution-stable (6–8%); noisy improvements grow at coarser resolutions, from 14% at 2048 up to 39% at 64: input-side derivative features are most valuable when the network's Nyquist headroom is tightest.

*Table 11.* Stress-test: relative $L^2$ error under Gaussian and uniform additive noise ($\sigma$ from 0.05 to 0.30), two backbones (FNO, LocalNO). Bold = best per (row, noise type, backbone); $\Delta_\partial$ in green. $\partial$-FNO improves the baseline in **60/60** settings; $\partial$-LocalNO improves in **60/60** settings; combined **120/120** wins. Peak improvement: 48.5% (Gaussian) and 48.1% (uniform) on smooth Burgers. $n$=1000 training samples. $\Delta_\partial = (\text{BL} - \partial\text{-Model})/\text{BL} \times 100$.

| | | Gaussian | | | | | | Uniform | | | | | |
| | | FNO | | | LocalNO | | | FNO | | | LocalNO | | |
| | $\sigma$ | BL | $\partial$ | $\Delta_\partial$ | BL | $\partial$ | $\Delta_\partial$ | BL | $\partial$ | $\Delta_\partial$ | BL | $\partial$ | $\Delta_\partial$ |
|---|---|---|---|---|---|---|---|---|---|---|---|---|---|
| **Burg. $\nu$=0.1** | 0.05 | 0.00259 | **0.00210** | 19.1% | 0.00261 | **0.00215** | 17.7% | 0.00138 | **0.00111** | 19.9% | 0.00137 | **0.00113** | 17.4% |
| | 0.10 | 0.00621 | **0.00459** | 26.0% | 0.00726 | **0.00518** | 27.3% | 0.00281 | **0.00206** | 26.9% | 0.00568 | **0.00348** | 38.8% |
| | 0.15 | 0.00944 | **0.00711** | 24.7% | 0.01264 | **0.00995** | 21.3% | 0.00530 | **0.00392** | 26.0% | 0.00908 | **0.00539** | 40.6% |
| | 0.20 | 0.01270 | **0.01050** | 17.4% | 0.01925 | **0.01584** | 17.7% | 0.00770 | **0.00598** | 22.3% | 0.01075 | **0.00757** | 29.6% |
| | 0.25 | 0.01782 | **0.01236** | 30.6% | 0.02295 | **0.01646** | 27.4% | 0.00943 | **0.00905** | 4.0% | 0.01540 | **0.00939** | 39.0% |
| | 0.30 | 0.02212 | **0.01493** | 32.5% | 0.03055 | **0.01573** | 48.5% | 0.01385 | **0.01188** | 14.2% | 0.02409 | **0.01250** | 48.1% |
| **Burg. $\nu$=10$^{-3}$** | 0.05 | 0.02593 | **0.02448** | 5.6% | 0.01697 | **0.01678** | 1.1% | 0.01519 | **0.01294** | 14.8% | 0.01034 | **0.01001** | 3.2% |
| | 0.10 | 0.03223 | **0.03150** | 2.2% | 0.02809 | **0.02570** | 8.5% | 0.01963 | **0.01757** | 10.5% | 0.01859 | **0.01504** | 19.1% |
| | 0.15 | 0.03276 | **0.03024** | 7.7% | 0.04088 | **0.03336** | 18.4% | 0.02407 | **0.02221** | 7.7% | 0.02646 | **0.02511** | 5.1% |
| | 0.20 | 0.03897 | **0.03630** | 6.9% | 0.05948 | **0.04410** | 25.9% | 0.02827 | **0.02683** | 5.1% | 0.03645 | **0.03363** | 7.7% |
| | 0.25 | 0.05195 | **0.05026** | 3.3% | 0.06839 | **0.05614** | 17.9% | 0.03248 | **0.03043** | 6.3% | 0.04087 | **0.03334** | 18.4% |
| | 0.30 | 0.04982 | **0.04644** | 6.8% | 0.07652 | **0.05710** | 25.4% | 0.03662 | **0.03444** | 5.9% | 0.05561 | **0.04098** | 26.3% |
| **KdV** | 0.05 | 0.02640 | **0.02567** | 2.8% | 0.04840 | **0.04597** | 5.0% | 0.01997 | **0.01974** | 1.2% | 0.03434 | **0.03238** | 5.7% |
| | 0.10 | 0.03575 | **0.03468** | 3.0% | 0.05649 | **0.05474** | 3.1% | 0.02447 | **0.02406** | 1.7% | 0.03874 | **0.03672** | 5.2% |
| | 0.15 | 0.04332 | **0.04230** | 2.3% | 0.05716 | **0.05524** | 3.4% | 0.03064 | **0.03038** | 0.8% | 0.04542 | **0.04184** | 7.9% |
| | 0.20 | 0.05590 | **0.05205** | 6.9% | 0.07070 | **0.07012** | 0.8% | 0.03722 | **0.03587** | 3.6% | 0.05419 | **0.04733** | 12.7% |
| | 0.25 | 0.06838 | **0.06455** | 5.6% | 0.08954 | **0.08492** | 5.2% | 0.04369 | **0.04222** | 3.4% | 0.06054 | **0.05383** | 11.1% |
| | 0.30 | 0.08012 | **0.06979** | 12.9% | 0.09959 | **0.08592** | 13.7% | 0.05126 | **0.04844** | 5.5% | 0.07272 | **0.05854** | 19.5% |
| **Darcy** | 0.05 | 0.05660 | **0.05430** | 4.2% | 0.05581 | **0.05529** | 0.9% | 0.05540 | **0.05456** | 1.5% | 0.05567 | **0.05306** | 4.7% |
| | 0.10 | 0.05644 | **0.05431** | 3.8% | 0.05583 | **0.05483** | 1.8% | 0.05643 | **0.05428** | 3.8% | 0.05560 | **0.05371** | 3.4% |
| | 0.15 | 0.05636 | **0.05476** | 2.8% | 0.05597 | **0.05483** | 2.0% | 0.05646 | **0.05435** | 3.8% | 0.05556 | **0.05371** | 3.3% |
| | 0.20 | 0.05646 | **0.05515** | 2.3% | 0.05625 | **0.05587** | 0.7% | 0.05666 | **0.05439** | 4.0% | 0.05574 | **0.05374** | 3.4% |
| | 0.25 | 0.05682 | **0.05582** | 1.7% | 0.05730 | **0.05652** | 1.4% | 0.05614 | **0.05438** | 3.1% | 0.05590 | **0.05409** | 3.2% |
| | 0.30 | 0.05744 | **0.05618** | 2.2% | 0.05744 | **0.05629** | 2.0% | 0.05667 | **0.05551** | 2.0% | 0.05549 | **0.05441** | 1.9% |
| **NS $\nu$=10$^{-5}$** | 0.05 | 0.03718 | **0.03631** | 2.3% | 0.02667 | **0.02472** | 7.3% | 0.03566 | **0.03513** | 1.5% | 0.02438 | **0.02279** | 6.5% |
| | 0.10 | 0.03861 | **0.03749** | 2.9% | 0.03147 | **0.02919** | 7.2% | 0.03659 | **0.03645** | 0.4% | 0.02745 | **0.02547** | 7.2% |
| | 0.15 | 0.04284 | **0.04115** | 4.0% | 0.03548 | **0.03309** | 6.7% | 0.03856 | **0.03807** | 1.3% | 0.02994 | **0.02810** | 6.2% |
| | 0.20 | 0.04435 | **0.04353** | 1.9% | 0.03879 | **0.03639** | 6.2% | 0.03925 | **0.03890** | 0.9% | 0.03286 | **0.03046** | 7.3% |
| | 0.25 | 0.04782 | **0.04419** | 7.6% | 0.04161 | **0.03949** | 5.1% | 0.04082 | **0.04045** | 0.9% | 0.03461 | **0.03241** | 6.4% |
| | 0.30 | 0.05150 | **0.04956** | 3.8% | 0.04444 | **0.04217** | 5.1% | 0.04211 | **0.04181** | 0.7% | 0.03700 | **0.03449** | 6.8% |

A practical implication is that downsampling, the most common strategy when GPU memory is limited, silently makes exact physics enforcement increasingly suboptimal. $\partial$-NO compensates automatically by learning lower orders at coarser resolutions, while fixed-order approaches cannot.

**L.5. Directional Isotropy on 2D Isotropic PDEs**

On PDEs whose differential operator is isotropic in space (no preferred axis), we expect $\beta_x \approx \beta_y$ to emerge if $\partial$-NO captures the true differential structure rather than incidental data features. We test this on Darcy (isotropic elliptic) and 2D Navier-Stokes ($\nu$=10$^{-5}$) with $K$=4 learnable channels.

Across both benchmarks, $\beta_x$ and $\beta_y$ converge within $\Delta < 0.035$ of each other without isotropy being prescribed. The channels self-organize into a hierarchy of spectral scales rather than collapsing to identical orders, indicating that the network uses the $K$=4 budget productively to span low through high frequencies.

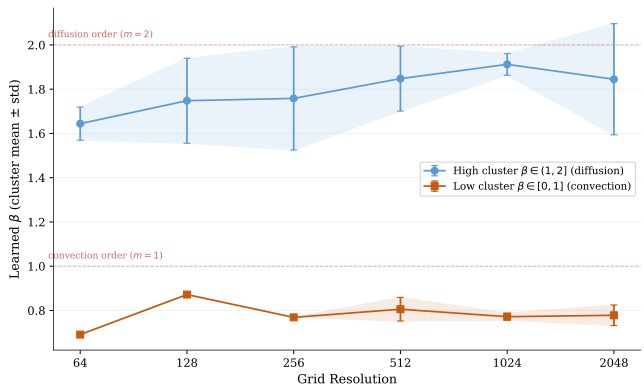

*Figure 5.* Learned $\beta$ clusters across resolutions (Burgers, $K{=}8$ learnable channels). Two clusters consistently emerge near $m{=}2$ (diffusion, top) and $m{=}1$ (convection, bottom). The diffusion cluster is *not* resolution-invariant: it drifts below 2 at coarser resolutions, an empirical effect of finite stencil fidelity at large grid spacing rather than a direct consequence of Theorem 5.6, whose closed form governs the SNR and capacity dependence of $\delta^*$. Downsampling, the most common strategy when GPU memory is limited, silently makes exact physics enforcement increasingly suboptimal. $\partial$-NO compensates automatically by learning lower orders.

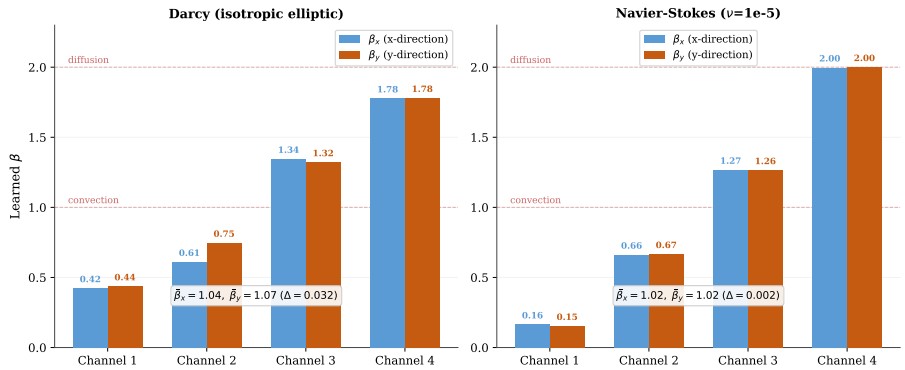

*Figure 6.* Directional isotropy of learned orders on two 2D isotropic PDEs. Left: Darcy. Right: Navier-Stokes ($\nu{=}10^{-5}$). For each of $K{=}4$ learnable channels, $\beta_x$ (blue) and $\beta_y$ (orange) converge to nearly identical values, with $\bar{\beta}_x{=}1.04$, $\bar{\beta}_y{=}1.07$ on Darcy ($\Delta{=}0.032$) and $\bar{\beta}_x{=}1.02$, $\bar{\beta}_y{=}1.02$ on NS ($\Delta{=}0.002$). The four channels self-organize into a hierarchy spanning $[0.15, 2.0]$, each capturing a different spectral scale. Isotropy is recovered without being prescribed.

## L.6. Three-Dimensional Compressible Navier-Stokes

We test $\partial$-augmentation on the 3D compressible Navier-Stokes equations from the PDEBench benchmark suite (Takamoto et al., 2022) ($64 \times 64 \times 64$ spatial grid, $T{=}10$, five-variable input $V_x, V_y, V_z$, density, pressure), with $K_x{=}K_y{=}K_z{=}2$ axis-aligned channels and $K_{xy}{=}K_{xz}{=}K_{yz}{=}K_{xyz}{=}1$ cross- and triple-derivative channels.

The axis-aligned channels learn orders close to the Laplacian order $m{=}2$, while cross- and triple-derivative channels saturate near identity. The directional structure is consistent across three backbones and persists under noise, supporting the claim that learned $\beta$ values reflect physically meaningful differential structure rather than backbone-specific quirks.

## L.7. Three-Dimensional Compressible Navier-Stokes: Quantitative Results

Complementing the learned-orders breakdown in §L.6, we report the quantitative test error on the 3D compressible Navier-Stokes benchmark ($64 \times 64 \times 64$, $T{=}10$, five-variable input). Three backbones (FNO3D, TFNO, Transolver) are evaluated under clean and noisy inputs; $\partial$-augmentation uses $K_x{=}K_y{=}K_z{=}2$ axis channels and $K_{xy}{=}K_{xz}{=}K_{yz}{=}K_{xyz}{=}1$ cross- and triple-derivative channels.

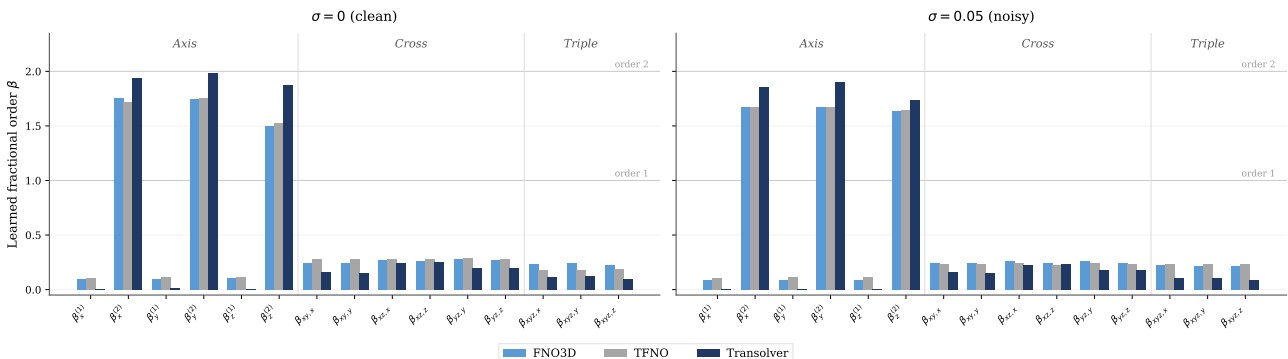

*Figure 7.* Learned fractional orders on 3D compressible Navier-Stokes, for three backbones (FNO3D, TFNO, Transolver) under clean ($\sigma$=0, left) and noisy ($\sigma$=0.05, right) conditions. Per-axis channels ($\beta_x, \beta_y, \beta_z$) learn orders near 2 (Laplacian, the dominant differential operator), while cross-derivative ($\beta_{xy}, \beta_{xz}, \beta_{yz}$) and triple-derivative ($\beta_{xyz}$) channels collapse to near-identity ($\beta \approx 0$). The network selectively amplifies the physically dominant differential structure without it being prescribed.

*Table 12.* 3D Compressible Navier-Stokes: test relative $L^2$ error. $\partial$-augmentation improves every (backbone, noise) cell; gains widen under noise on Transolver.

|  | **FNO3D** | | **TFNO** | | **Transolver** | |
|---|---|---|---|---|---|---|
|  | $\sigma=0$ | $\sigma=0.05$ | $\sigma=0$ | $\sigma=0.05$ | $\sigma=0$ | $\sigma=0.05$ |
| Model | 0.2101 | 0.2168 | 0.2079 | 0.2225 | 0.1990 | 0.2039 |
| $\partial$-Model | **0.1964** | **0.2115** | **0.1971** | **0.2033** | **0.1699** | **0.1695** |
| *Improv.* | 6.5% | 2.4% | 5.2% | 8.6% | 14.6% | 16.9% |

Transolver gains are largest both clean and noisy, consistent with its larger effective capacity and the bias-variance prediction that higher-capacity models benefit more from a regularization-aware input feature.

### L.8. Conv-FNO Standalone Comparison

We compare $\partial$-FNO against Conv-FNO (Liu et al., 2025) as direct alternatives: both preprocess the input, $\partial$-FNO with a GL derivative stencil and Conv-FNO with learnable convolution kernels. Both build on the same FNO backbone, allowing a clean head-to-head test of preprocessing strategies.

*Table 13.* Conv-FNO vs. $\partial$-FNO on three benchmarks. Relative $L^2$ error (%); $\Delta$ is the relative improvement over the FNO baseline. $\partial$-FNO matches or outperforms Conv-FNO with **8–128$\times$ fewer** preprocessing parameters (see note below).

|  |  |  |  |  | *Improv. over FNO* | |
|---|---|---|---|---|---|---|
| **Dataset** | $\sigma$ | **FNO** | $\partial$**-FNO** | **Conv-FNO** | $\partial$**-FNO** | **Conv-FNO** |
| Burgers ($\nu$=0.1) | 0 | 0.0832 | **0.0738** | 0.0791 | 11.3% | 4.9% |
| Burgers ($\nu$=0.1) | 0.02 | 0.1161 | **0.0899** | 0.1052 | 22.6% | 9.4% |
| Darcy | 0 | 5.736 | **5.474** | 5.687 | 4.6% | 0.9% |
| Darcy | 0.02 | 5.721 | **5.487** | 5.678 | 4.1% | 0.8% |
| NS ($\nu$=$10^{-5}$) | 0 | 3.506 | **3.456** | 4.219 | 1.5% | $-20.3\%$ |
| NS ($\nu$=$10^{-5}$) | 0.02 | 3.614 | **3.575** | 4.208 | 1.1% | $-16.4\%$ |

*Parameter counts.* On Burgers, $\partial$-FNO adds $\approx 16$ preprocessing parameters (GL stencils for two channels); Conv-FNO adds $\approx 136$ (16-wide kernels per channel). On Darcy/NS, $\partial$-FNO adds $\approx 16$ vs Conv-FNO's $\approx 2056$. The constrained inductive bias matches or beats the learned kernel at 8–128$\times$ lower parameter overhead.

**L.9. Ablations on $K$ and $K_{\mathrm{GL}}$**

Two architectural hyperparameters control the cost-quality tradeoff of $\partial$-augmentation: the number of learnable derivative channels $K$ and the length of the Grünwald-Letnikov stencil $K_{\mathrm{GL}}$. We ablate both on Burgers ($\nu=0.1$) under clean ($\sigma=0$) and noisy ($\sigma=0.02$) conditions.

**(a)  Effect of Number of Derivative Channels $K$**

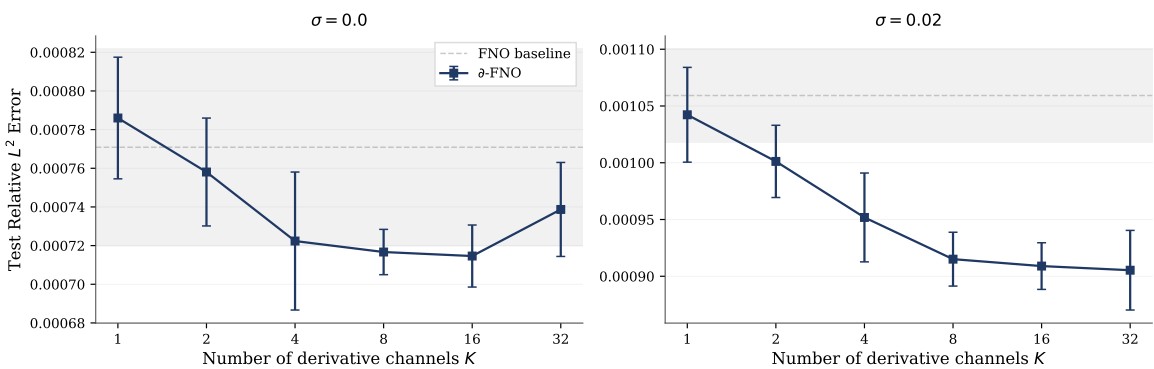

*Figure 8.* Ablation on the number of derivative channels $K \in \{1, 2, 4, 8, 16, 32\}$. Increasing $K$ consistently improves performance with diminishing returns past $K=8$. The natural tradeoff is memory and compute cost. Under noise the curve flattens more slowly, indicating that additional channels help more when the regularization burden is larger.

**(b)  Sensitivity to GL Stencil Length $K_{\mathrm{GL}}$**

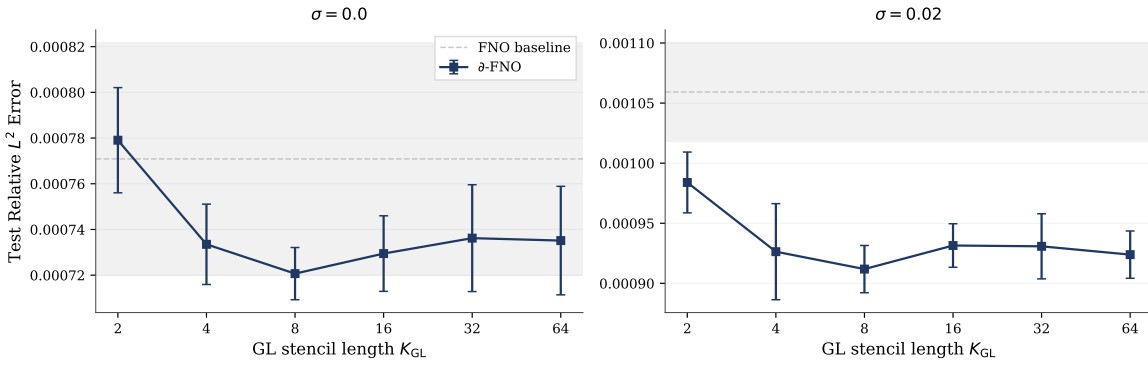

*Figure 9.* Sensitivity to GL stencil length $K_{\mathrm{GL}} \in \{2, 4, 8, 16, 32, 64\}$. Performance stabilizes around $K_{\mathrm{GL}}=16$ under both clean and noisy conditions. At $K_{\mathrm{GL}}=2$, accuracy degrades because the stencil is too short to resolve non-integer orders. We use $K_{\mathrm{GL}}=16$ throughout the main paper as a balance between accuracy and compute.

Both ablations support the default choices used in the main paper ($K_{\mathrm{GL}}=16$ universally). Increasing $K$ further would improve accuracy at proportional memory cost; reducing $K_{\mathrm{GL}}$ below 16 would save compute at the cost of resolution-dependent degradation of fractional orders.

**L.10. Trajectory of Learned $\beta$ During Training**

Theorem 5.6 characterizes the population-optimal $\beta^*$, but in practice we recover $\beta$ via stochastic gradient descent. Figure 10 traces all $K=8$ learnable orders across training epochs on Burgers ($\nu=0.1$), under five noise levels $\sigma \in \{0, 0.01, 0.02, 0.05, 0.1\}$.

Three takeaways. First, all eight orders stabilize within 200–400 epochs, indicating optimization is reliable. Second, the final distribution is noise-dependent: at $\sigma=0$ orders cluster near $\beta \in [0.5, 2.0]$, while at $\sigma=0.1$ the cluster shifts toward $[0.3, 1.8]$, exactly the direction Theorem 5.6 predicts. Third, $\beta$ values redistribute across the channel budget rather than

**(c) Evolution of Learned Derivative Orders During Training** (Burgers, $\nu = 0.1$, $K = 8$)

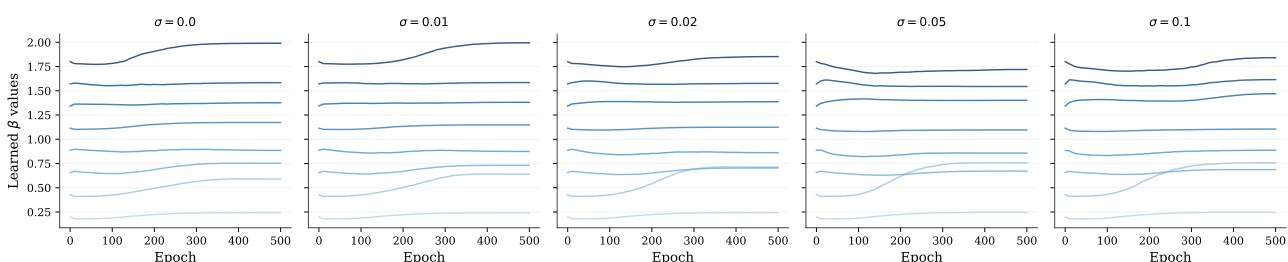

*Figure 10.* Trajectory of learned $\beta$ values during training (5 noise levels). All $K{=}8$ orders stabilize within 200–400 epochs and converge to values consistent with Theorem 5.6: higher $\beta^*$ under clean data, systematically lower $\beta^*$ as $\sigma$ grows. Initial values are $\beta_1{=}1.0$ and $\beta_2{=}2.0$; the optimizer redistributes channels across the spectral hierarchy as training proceeds.

collapsing to redundant orders, validating the design choice of multiple learnable channels per direction.

## L.11. Interaction Between Noise Level and Channel Count

The K-ablation (Fig. 8) and the stress test (Table 11) study the effects of channel count $K$ and noise level $\sigma$ separately. Figure 11 shows their interaction: the relative improvement of $\partial$-FNO over FNO as a heatmap over the joint $(\sigma, K)$ plane on Burgers ($\nu{=}0.1$). The super-additivity has a clean theoretical reading: each channel $\beta_k$ captures a different point on the

**Relative improvement of $\partial$-FNO over FNO (% reduction in rel. $L_2$)**

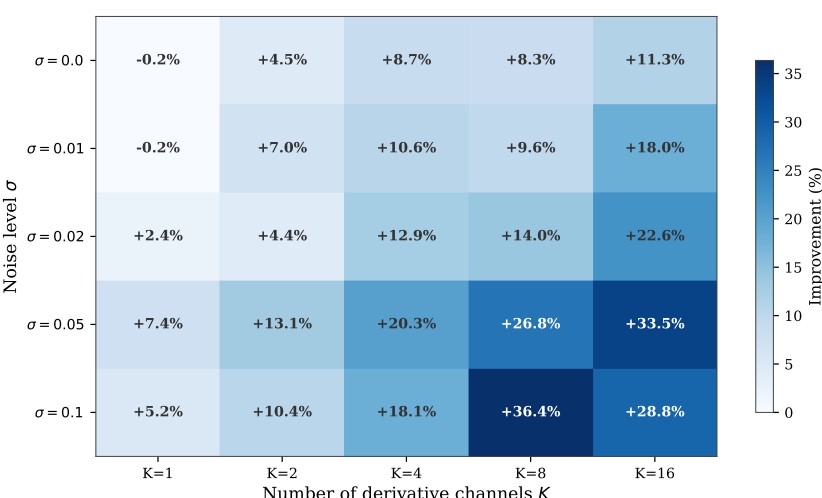

*Figure 11.* Interaction between noise level $\sigma$ and number of derivative channels $K$. Cell values are the relative improvement of $\partial$-FNO over FNO (% reduction in relative $L^2$) on Burgers ($\nu{=}0.1$). Two patterns: (i) at fixed $K$, gains grow with $\sigma$; (ii) at fixed $\sigma$, gains grow with $K$ with diminishing returns past $K{=}8$. The interaction is super-additive: the largest single gain ($+36.4\%$) sits at ($\sigma{=}0.1$, $K{=}8$), in the high-noise regime near the diminishing-returns plateau of Figure 8, supporting the design choice of multiple channels precisely when data is noisy.

bias-variance curve from Theorem 5.6, so when noise broadens the optimal $\beta$ distribution the diverse-channel budget pays off more than the sum of its parts.

