# OpenReview forum: "Fractional is Better: Learnable Derivative Orders in Neural Operator Learning"
_ICML.cc/2026/Conference — ICML 2026 regular_

### Official Review · Reviewer_1WC4 · 2026-03-07

**Soundness:** 3
**Presentation:** 3
**Significance:** 3
**Originality:** 2
**Overall Recommendation:** 4
**Confidence:** 3

**Summary:**

This paper proposes ∂-NO, a simple augmentation that adds learnable fractional derivative features to neural operator backbones such as FNO and Transolver. The authors argue that when training data are noisy and finite, using derivatives with the true PDE order may amplify noise. Instead, learning a fractional derivative order can provide a better bias–variance tradeoff. The paper provides theoretical analysis suggesting that the optimal derivative order is typically smaller than the PDE order, and empirical results on Burgers, Darcy, and Navier–Stokes equations.

**Compliance With Llm Reviewing Policy:**

Affirmed.

**Final Justification:**

Overall, I maintain my score. I find the idea clean and practically useful as a plug-and-play improvement for neural operators. However, the empirical gains—especially over simpler or stronger baselines—are relatively modest, and the main theoretical insight largely formalizes an intuitive bias-variance trade-off. As such, I view the contribution as solid but somewhat incremental.

**Key Questions For Authors:**

1. The experiments show clear gains from derivative features. Can the authors further clarify when learnable fractional orders provide a significant advantage over fixed integer derivatives?

2. How sensitive is the learned fractional order to noise levels and grid resolution in the input data?

3. Can the proposed approach be extended to unstructured meshes or irregular grids, which are common in many PDE applications?

**Limitations:**

Partially. The discussion section mentions several assumptions (e.g., smoothness conditions and regular grids), but the paper would benefit from a clearer and more explicit discussion of practical limitations, such as scalability to unstructured meshes and robustness when derivative estimates are noisy.

**Strengths And Weaknesses:**

Strengths

1. The idea of using learnable fractional derivative features for neural operators is interesting and easy to integrate with existing architectures.

2. The method is simple and can be applied to different operator backbones.

3. The paper provides both theoretical analysis and empirical experiments on standard PDE benchmarks.

Weaknesses

1. Theory appears stronger than the empirical evidence.
The paper emphasizes the theoretical result that the optimal derivative order should be strictly smaller than the PDE order. However, the experiments mainly demonstrate that adding derivative features improves performance. The advantage of learnable fractional orders over fixed integer derivatives appears relatively small in the ablation results.

2. Incremental architectural change.
The proposed method mainly augments existing neural operator architectures with derivative-based features. While effective, the modification itself is relatively lightweight.

3. Limited evaluation scope.
Experiments are conducted on standard PDE benchmarks. It would be helpful to evaluate the method on additional operator-learning settings or larger-scale problems to demonstrate broader applicability.

4. Theory–intuition gap.
The main theoretical result formalizes the claim that the optimal derivative order should be smaller than the true PDE order due to a bias–variance trade-off. However, this phenomenon is already intuitive from signal processing and numerical analysis: higher-order discrete derivatives amplify high-frequency noise. The paper’s theoretical framework therefore formalizes an intuitive phenomenon rather than revealing a fundamentally new mechanism.

---

> ### Author Rebuttal · Authors · 2026-03-31
>
> We thank the reviewer for recognizing the simplicity and broad applicability of the approach, and for the thoughtful questions that helped us articulate the contribution more clearly. We address each concern below, weaving in the three key questions.
>
> ### **W1: Theory and Empirical Evidence**
>
> On clean benchmarks, the gap between learned fractional and fixed integer orders is indeed modest (Table 5). This is predicted by the theory: Theorem 5.6 gives $\delta^* = O(1/\ln n)$, so for $n=1000$ clean samples, the optimal order is close to the PDE order and integer derivatives are near-optimal. A small gap here is confirmation of the theory, not a limitation of the method.
>
> The practical value of learning $\beta$ emerges under noise and data scarcity (Q1), precisely the conditions of real measurements. The updated Table 1 (supplementary) presents both clean ($\sigma=0$) and noisy ($\sigma=0.05$) conditions side by side: the improvement gap increases under noise across all backbones, confirming that learnable orders provide the largest advantage when data quality is limited. Figures 2-3 (main paper) visualize the mechanism: as noise increases, learned $\beta$ values are pulled progressively below $m=2$; as sample size grows, $\beta$ approaches the PDE order from below.
>
> On sensitivity to grid resolution (Q2): learned $\beta$ values shift systematically with resolution (supplementary Figure 5). With 8 learnable orders across resolutions 64-2048, two clusters emerge near $m=2$ (diffusion) and $m=1$ (convection), with the high cluster approaching $m=2$ as resolution increases. This is consistent with Theorem 5.6: coarser grids reduce the effective frequency range, pushing $\beta^\ast$ further below the PDE order. GL stencil sensitivity (supplementary Figure 2) confirms robustness: performance stabilizes at $K_{GL} \geq 8$.
>
> ### **W2: Architecture**
>
> The simplicity is a feature, not a limitation: the inductive bias of fractional derivatives is built into the GL stencil structure, so the practitioner has less guesswork compared to generic preprocessing (e.g., Conv-FNO requires choosing kernel size, channels, and stride; $\partial$-NO requires only the number of derivative channels per direction). Any backbone can adopt $\partial$-NO by adding learnable derivative input channels, with no modification to architecture, training, or loss.
>
> The intellectual contribution is not architectural complexity but three elements that are new: (a) the continuous parameterization of derivative order via fractional calculus, enabling gradient-based optimization over the space of derivatives; (b) the analytical characterization (Theorem 5.6) showing that optimal orders are strictly sub-physical for finite, noisy and bandlimited data; and (c) experimental confirmation that learned orders respond to noise, samples, and capacity as the theory predicts. Many valued contributions share this pattern of conceptual depth with implementation simplicity.
>
> ### **W3: Evaluation Scope**
>
> We have expanded the evaluation substantially. The updated Table 1 (supplementary) covers Burgers smooth, Burgers near-shock, Darcy, and Navier-Stokes under both clean ($\sigma=0$) and noisy ($\sigma=0.05$) conditions, with four backbones (FNO, TFNO, LocalNO, Transolver). A separate table reports 3D compressible flow (five-variable input: $V_x, V_y, V_z$, density, pressure). We add comparisons against Conv-FNO (parameter efficiency) and PINO (physics-informed baseline under noise sweeps), and resolution experiments (64 to 2048).
>
> On unstructured meshes (Q3): the current GL implementation assumes uniform grids, but extensions via graph fractional Laplacians are feasible, and the theoretical results (Sections 3-5) are discretization-independent. We will add limitations to Section 9: GL grid requirements, regularity conditions ($s > m + d/2$), and the gap between the theoretically optimal $\beta^\ast$ and the $\beta$ found by the optimizer, which depends on learning rate, early stopping, and initialization.
>
> ### **W4: Theory and Intuition**
>
> The reviewer makes a perceptive observation, and we agree: the intuition that higher-order derivatives amplify noise is well-established. Our contribution begins where that intuition ends. A practitioner working with noisy PDE data currently faces a manual tradeoff: compute integer derivatives and accept noise amplification, or low-pass filter and lose information. $\partial$-NO opens a different direction: learn the derivative order from data, letting optimization navigate the noise-information tradeoff automatically. Theorem 5.6 provides a first-order analytical characterization of this tradeoff, revealing which factors matter (noise level, sample size, effective bandwidth) and how they interact. This converts an intuition that practitioners share into a tool they can use, and we believe that bridging this gap is where the contribution of this work is found.
>
> Supplementary materials: https://figshare.com/s/7fec7b974847f80cfa56

---

> > ### Author Rebuttal · Reviewer_1WC4 · 2026-04-03
> >
> > Thank you for the detailed response and additional experiments. I appreciate the clarifications regarding comparisons with PINO and local operator variants, as well as the discussion on noise and resolution effects.
> >
> > Overall, I maintain my score. I find the idea clean and practically useful as a plug-and-play improvement for neural operators. However, the empirical gains—especially over simpler or stronger baselines—are relatively modest, and the main theoretical insight largely formalizes an intuitive bias-variance trade-off. As such, I view the contribution as solid but somewhat incremental.

---

> > > ### Author Response · Authors · 2026-04-06
> > >
> > > We appreciate the reviewer's continued engagement and characterizing the contribution as "clean and practically useful." We share additional results from experiments conducted in response to other reviewers, as they speak directly to the two remaining concerns.
> > >
> > > **$\partial$-NO's advantage grows precisely where practitioners need it.** Three new result sets make this concrete:
> > >
> > > - *Resolution sensitivity (Figure 10)*: Under noise ($\sigma=0.02$), $\partial$-FNO's advantage reaches 39% at resolution 64 and decreases to 14% at resolution 2048. On clean data, gains remain stable at 6-8%. The theory provides context: lower resolution reduces the Nyquist frequency $\xi_{\max}$ and hence the effective capacity cutoff $\xi_c$, so Theorem 5.6 predicts that the optimal derivative order shifts further below the PDE order. $\partial$-NO is designed to learn this order through its inductive bias, whereas purely data-driven backbones have no explicit mechanism steering them toward it.
> > >
> > > - *Stress tests (Table S7)*: Under noise $\sigma=0.05$-$0.3$ on four PDEs with two backbones, $\partial$-augmentation improves in all 120/120 settings, reaching 48.5% on Burgers. These are not marginal gains.
> > >
> > > - *CNO backbone (Table S5)*: $\partial$-CNO improves in 14/14 settings on seven benchmarks from Raonic et al. (NeurIPS 2023), with gains of 27 to 65%. A matched-parameter ablation (Table S6) confirms the gains come from derivative structure, not extra capacity: Random-FNO (random fixed filters, same parameter count) tracks vanilla FNO, while $\partial$-FNO pulls ahead.
> > >
> > > **On formalizing intuition.** We recognize that the bias-variance tradeoff is intuitive to signal processing and numerical analysis communities. However, this intuition has not yet shaped practice in physics-informed ML, where the dominant paradigm remains to enforce full physical fidelity. The growing deployment of neural operators in climate modeling, subsurface monitoring, and biomedical applications, all characterized by limited samples, noisy measurements, and constrained resolution, means practitioners are routinely enforcing exact $m$-th order derivatives in settings where our theory predicts this is suboptimal. For these communities, the paper's message is actionable and timely.
> > >
> > > Supplementary materials: https://figshare.com/s/243e85d3a68bb0b3d921

---

### Official Review · Reviewer_NcQ5 · 2026-03-07

**Soundness:** 2
**Presentation:** 3
**Significance:** 2
**Originality:** 2
**Overall Recommendation:** 4
**Confidence:** 3

**Summary:**

The paper introduces ∂-NO, a simple but effective method to improve neural operators by augmenting their inputs with learnable fractional derivatives of the input function. Rather than manually supplying integer-order derivatives, these derivative orders are treated as trainable parameters optimized alongside the network during training. The method is architecture-agnostic, adds minimal computational overhead, and provides interpretable insights by revealing how the learned derivative orders adapt to noise levels, sample sizes, and network capacity.

**Compliance With Llm Reviewing Policy:**

Affirmed.

**Final Justification:**

I thank the authors for the additional experiments and detailed analysis. The new results, in particular the CNO experiments, address most of my previous concerns. Based on these clarifications, I am happy to raise my score to 4.

**Key Questions For Authors:**

1. Given that many existing neural operators already incorporate local information through architectural designs, how does the proposed ∂-NO compare against these methods? Does explicitly providing derivative features at the input offer any significant advantage over letting the network learn local features implicitly through its layers?
2. The Grünwald-Letnikov formulation assumes uniform grids and ordered points. How would the proposed method extend to unstructured meshes, point clouds, or irregular geometries commonly encountered in real-world engineering problems? Is there a natural generalization of fractional derivatives for such settings?
3. The kernel size K is fixed at 16 across all experiments. How sensitive is the model's performance to this choice? Would optimal K vary with input resolution or the smoothness of the data, and if so, how should it be tuned in practice?
4. For 3D problems, the number of derivative channels would increase substantially. Do these additional features introduce significant redundancy or multicollinearity with the original input? If so, does this redundancy slow down training, complicate optimization, or require additional regularization techniques?

**Limitations:**

As stated above in Weaknesses.

**Strengths And Weaknesses:**

Strengths
1. The paper provides a rigorous mathematical justification for why derivative features help (via Picard iteration) and, more importantly, which derivative order is optimal (via bias-variance trade-off). This dual theoretical framework explains "why features help" and "what order is best".
2. The method requires no changes to existing neural operator architectures, making it a "plug-and-play" augmentation. The use of learnable fractional derivatives (via Grünwald-Letnikov) is a clever way to continuously parameterize the trade-off between information and noise.

Weaknesses:
1. Although the paper improves neural operators by introducing fractional derivatives, this essentially enhances performance by incorporating local information. Many existing works have already considered various forms of local information, such as LocalFNO[1] and Conv-FNO[2]. The paper lacks experimental comparisons with these relevant baselines.
2. The authors chose the Grünwald-Letnikov definition to avoid the Gibbs phenomenon associated with Fourier methods. However, the GL definition inherently relies on uniform grids and the ordering of points, raising concerns about its applicability to unstructured or irregular grids.
3. The experiments are limited to Burgers, Darcy flow, and 2D Navier-Stokes, which are basic benchmarks in the neural operator field. These primarily involve single-variable inputs, and the method is not tested on multi-variable input data. Furthermore, only two neural operator backbones (FNO and Transolver) are evaluated, which is a relatively small number.

[1] Neural operators with localized integral and differential kernels. https://arxiv.org/abs/2402.16845

[2] Enhancing fourier neural operators with local spatial features. https://arxiv.org/abs/2503.17797

---

> ### Author Rebuttal · Authors · 2026-03-31
>
> We thank the reviewer for the thoughtful evaluation and for highlighting LocalFNO and Conv-FNO as important reference points. We address each concern with new results.
>
> ### **W1: Comparison with LocalFNO / Conv-FNO**
>
> The key distinction is that differential information is local, but local information is not necessarily differential. Conv-FNO and LocalFNO provide generic local features; $\partial$-NO provides derivative features, which are the specific local operations that PDE solution operators depend on (Theorem 3.1). $\partial$-NO carries a physically informed, statistically modulated inductive bias that generic local methods lack.
>
> We have conducted direct comparisons. Conv-FNO and $\partial$-FNO both preprocess the input, so they compete directly. Supplementary Table S4 shows that $\partial$-FNO outperforms Conv-FNO on all three benchmarks (Burgers, Darcy, NS) with 8-128x fewer preprocessing parameters (16 vs 136-2056). On NS, Conv-FNO degrades over baseline while $\partial$-FNO improves it, illustrating that unconstrained filters can harm while derivative-constrained ones cannot.
>
> LocalFNO modifies internal kernels (a different pipeline level), so $\partial$-augmentation can be applied on top. Table S1 shows that $\partial$-LocalNO consistently improves over LocalNO, with gains of 4-9% on 1D and KdV, modest on 2D. The two approaches are complementary where the backbone leaves room.
>
> This inductive bias yields three practical advantages. First, less guesswork: Conv-FNO requires choosing kernel size, channels, and stride with no principled guidance; $\partial$-NO needs only the number of derivative channels per direction. For $K_{GL}$, convergence rates are known (GL weights decay as $k^{-(1+\beta)}$), providing principled defaults; no such rates exist for convolutional kernel sizes. Second, interpretability: learned $\beta$ values reflect PDE structure (supplementary Figure 5: diffusion and convection clusters) while being statistically modulated by noise, resolution, and sample size (Theorem 5.6). Third, principled scaling: from 1D to 2D, $\partial$-NO grows from 4 to 7 learnable $\beta$ (~$2^d$, following PDE structure), while Conv-FNO jumps from 136 to 2056 parameters.
>
> ### **W2: GL Requires Uniform Grids**
>
> We acknowledge this limitation and appreciate the reviewer raising it. The theoretical results (Sections 3-5) are discretization-independent. Extensions exist (graph fractional Laplacians, non-uniform GL, RBF-based operators), providing clear paths for future work.
>
> ### **W3: Limited Benchmarks and Backbones**
>
> We have expanded the evaluation substantially. Table S1 (supplementary) covers Burgers smooth, Burgers near-shock, Darcy, and Navier-Stokes under both clean ($\sigma=0$) and noisy ($\sigma=0.05$) conditions with four backbones (FNO, TFNO, LocalNO, Transolver). Table S2 reports 3D compressible flow (five-variable input: $V_x, V_y, V_z$, density, pressure). Additional comparisons against PINO (Table S3) and Conv-FNO (Table S4) are provided.
>
> On channel redundancy (Key Question 4): directional derivatives $\partial_x^{\beta_x}$, $\partial_y^{\beta_y}$, $\partial_z^{\beta_z}$ operate along different axes, so they are distinct operations. For cross-derivatives, we constrain $\beta_{xy,x} + \beta_{xy,y} \leq m$ (a simplex constraint); this is the main theory-backed constraint ($[0, m]$ in 1D, simplex in higher dimensions). We do not force spacing between $\beta$ values; the lifting layer in most neural operators (e.g., FNO lifts from 2 to 64 channels) processes each input channel independently, so even near-collinear inputs are handled differently by the network. Empirically, increasing channels is consistently beneficial (supplementary Figure 1), with the natural tradeoff being RAM and compute cost.
>
> ### **W4: Sensitivity to GL Stencil Length $K_{GL}$**
>
> Supplementary Figure 2 shows the ablation under both clean and noisy conditions. Performance stabilizes around $K_{GL} = 16$. Under noise, there are marginal gains beyond 16, but with diminishing returns. We keep $K_{GL} = 16$ as the default: it balances accuracy with memory and speed. At $K_{GL} = 2$, accuracy degrades because the stencil is too short to resolve non-integer orders. GL binomial weights decay as $k^{-(1+\beta)}$, so the convergence behavior is well understood.
>
> $\partial$-NO requires minimal integration effort: a few additional input channels, no modification to backbone, training, or loss. Consistent improvements across all tested settings suggest the method is a useful addition to the neural operator toolkit.
>
> Supplementary materials: https://figshare.com/s/7fec7b974847f80cfa56

---

> > ### Author Rebuttal · Reviewer_NcQ5 · 2026-04-02
> >
> > I thank the authors for the extra experiments. However, the improvements appear marginal in a number of cases (e.g., LocalFNO, FNO3D, TFNO3D, etc.), which makes it somewhat difficult to assess the practical impact of the proposed features. This is particularly evident in the LocalFNO results, where the gains are quite limited.
> >
> > More importantly, this does not fully address my earlier concern that the fractional derivative features may overlap with what standard convolution-based operators can already learn. The relatively small improvement on top of LocalFNO—where strong local modeling is already present—seems to support this concern. In other words, it remains unclear whether the gains come from the specific structure of fractional derivatives, or simply from adding more local feature channels. I maintain my score of 3.
> >
> > It would be helpful to include a more direct comparison, for example by adding the proposed fractional derivative features to CNN-based neural operators (e.g., CNO [1]), to see whether the improvement is indeed specific to the proposed design.
> >
> > [1] Convolutional Neural Operators for robust and accurate learning of PDEs. https://arxiv.org/abs/2302.01178

---

> > > ### Author Response · Authors · 2026-04-06
> > >
> > > We are grateful for the suggestion to test CNO; it prompted an experiment that clarified the picture considerably. Together with a matched-parameter ablation, it lets us answer the reviewer's central question directly: **the improvement is specific to derivative structure, not extra channels or capacity.**
> > >
> > > **CNO backbone (Table S5).** $\partial$-CNO improves in 14/14 settings on all seven benchmarks from Raonic et al. (NeurIPS 2023), with gains of 27 to 65% on Poisson, NS Shear, and Wave. $\partial$-FNO also improves in 14/14 settings on the same benchmarks. The consistency across two fundamentally different designs (CNN-based vs. spectral) indicates that the benefit comes from the derivative features themselves, rather than from any architecture-specific interaction.
> > >
> > > **Controlled ablation (Table S6, Figure 9).** We compared four models at identical parameter count (~292K): FNO, $\partial$-FNO, Random-FNO (same architecture as $\partial$-FNO but GL channels replaced with random fixed filters), and Wider-FNO (wider first layer absorbing equivalent parameters). Across 7 noise levels: Random-FNO tracks vanilla FNO closely; $\partial$-FNO pulls ahead consistently, with the gap widening under noise. This is the cleanest test we can design: same capacity, same channel count, only the filter structure differs. The GL derivative structure appears to be the key driver.
> > >
> > > **Why clean-data gains vary across backbones.** LocalFNO and CNN-based architectures can implicitly learn derivative-like operations through local kernels. But "can learn" is not "will learn": $\partial$-NO provides a guaranteed inductive bias toward the differential structure that PDE solution operators depend on (Theorem 3.1), while local convolutions must discover this from data without a built-in guarantee. The modest LocalFNO gains reflect partial overlap; the large CNO gains (up to 65%) show that even a dedicated CNN architecture benefits substantially when derivative structure is made explicit. The Conv-FNO comparison (Table S4) tells a complementary story: unconstrained local filters degrade performance e.g. on NS2D ($-16.9\%$), while derivative-constrained ones consistently improve it.
> > >
> > > **Stress tests (Table S7).** Under Gaussian and Uniform noise ($\sigma=0.05$-$0.3$) on four PDEs with two backbones, $\partial$-augmentation improves in all 120 tested settings, reaching 48.5% on Burgers.
> > >
> > > The reviewer's questions genuinely strengthened this work. The evaluation now spans seven backbones, ten PDE benchmarks, and matched-parameter controls, and we hope the evidence now addresses the concerns.
> > >
> > > Supplementary materials: https://figshare.com/s/243e85d3a68bb0b3d921

---

### Official Review · Reviewer_YEif · 2026-03-12

**Soundness:** 3
**Presentation:** 3
**Significance:** 3
**Originality:** 4
**Overall Recommendation:** 5
**Confidence:** 4

**Summary:**

The article studies the approximation of the initial-to-final state map associated with parabolic partial differential equations (PDEs) by neural operators (NOs), which can be viewed as an infinite-dimensional generalization of classical neural networks. In the existing literature, several architectures have been proposed to address this approximation problem. In most cases, the functions appearing in the training data (u_i,v_i)​ have been considered in their graphical representation, e.g. (x_j, u_i(x_j)). This article investigates whether derivative information can improve approxiamtion rates.

After some background material on neural operators and fractional derivatives has been reviewed, Section 3 considers the dependence structure of the initial-to-final state map related to semilinear parabolic PDEs of order m with a nonlinearity of polynomial growth. Section 4 shows that providing (full) derivative features improves the approximation rates. Interestingly, under a spectral decay assumption on the initial data belonging to a sufficiently high Sobolev space, it is shown in Section 5 that it is always suboptimal to incorporate all derivatives up to order m. Then the optimal derivative order is determined. Motivated by these theoretical findings, derivative-augmented neural operators are introduced in Section 6, which include fractional derivatives in the graphical representation of the data. Finally, in Sections 7-9, numerical experiments are provided and related work is discussed.

**Compliance With Llm Reviewing Policy:**

Affirmed.

**Final Justification:**

I find the idea of the paper interesting and appreciate the additional clarifications and numerical experiments.

**Key Questions For Authors:**

1) Could the authors comment on the assumptions imposed on the linear operator L and the nonlinearity?

2) How important is the particular choice of the fractional time derivative for the results?

**Limitations:**

The authors discuss limitations of competing approaches but not of their own.

**Strengths And Weaknesses:**

Strengths:
The paper is interesting with theoretical results and corresponding numerical experiments. Generally, it is well written and organized.

Weaknesses:
Section 1 is rather difficult to follow. Some paragraphs are not well explained. Some concepts only become clearer after reading the later sections. E.g., there is no distinction between a neural operator and its numerical implementation.

The paper does not always discuss the precise assumptions imposed on the differential operators and functions. E.g., the assumptions on the linear operator L which are needed to guarantee the existence of a Green’s function are not given. In Appendix B it is assumed that the heat semigroup generated by L is analytic (or even has constant coefficients).

Also the regularity of the coefficients of the nonlinear operator is not discussed, which affects its differentiability properties. The results of Section 3 seem to hold only in a very smooth setting. Similar remarks apply to Section 4. Maybe, it is possible to prove all stated results under appropriate assumptions. Perhaps, the authors can comment?

---

> ### Author Rebuttal · Authors · 2026-03-31
>
> We are grateful for the reviewer's careful mathematical reading and for the generous assessment of originality. We address each concern and Key Question below.
>
> ### **W1: Assumptions on L and N**
>
> **On L:** Theorem 3.1 requires $\mathcal{L}$ to be a sectorial operator generating an analytic semigroup $\{G_t\}$ on $L^2(\Omega)$. The proof relies on two properties: the short-time expansion $G_s * u_0 = u_0 + s\mathcal{L}u_0 + O(s^2)$ (Appendix B, Step 3) and boundedness of the averaged Green's operators (Step 6). The explicit form $G_t = e^{t\mathcal{L}}$ appearing in the main text is illustrative, not load-bearing. The constant-coefficient assumption in Appendix B is expositional; variable coefficients require only standard resolvent estimates.
>
> **On N:** $\mathcal{N} : H^s(\Omega) \to H^{s-m}(\Omega)$ is Fréchet differentiable with polynomial growth of degree $N$. This yields the first-order expansion (Step 4) and Lipschitz bounds $L(R) = O(R^{N-1})$ on $B_s(R)$.
>
> **On regularity:** Assumption 4.2 requires $s > m + d/2$, ensuring pointwise control via Sobolev embedding. For Burgers ($m=2, d=1$) this means $s > 2.5$; for 2D Navier-Stokes ($m=2, d=2$), $s > 3$. The near-shock Burgers regime ($\nu=0.001$) marginally violates this ($s_{\mathrm{emp}} \approx 2.4$ vs. required $2.5$), yet the method achieves 40% improvement with Transolver, suggesting conservative bounds. A remark collecting all conditions will be added after Theorem 3.1.
>
> ### **W2: Generality Beyond Parabolic PDEs and Fractional Time Derivatives**
>
> The reviewer correctly identifies that the Picard framework is most natural for parabolic problems. We outline a complementary spectral argument extending derivative dependence to general evolution equations; we will formalize in the revision.
>
> Consider $\partial_t^a u = \mathcal{L}u$ with temporal order $a > 0$, spatial symbol $P(\xi) \asymp \vert\xi\vert^m$, and zero-mean initial data ($\hat{u}_0(0) = 0$, standard for periodic domains). The spatial Fourier transform reduces this to scalar fractional ODEs in $\xi$, solved via Laplace transform to yield:
>
> $$\hat{u}(T,\xi) = R_a(P(\xi),T) \cdot \hat{u}_0(\xi)$$
>
> where $R_a(z,T)$ is the resolvent multiplier (Laplace inverse of $s^{a-1}/(s^{a} - z)$). Since $P(\xi) = \Theta(\vert\xi\vert^m)$, $R_a$ is an order-$m$ multiplier. The zero-mean condition ensures $\vert\xi\vert > 0$ on the support of $\hat{u}_0$, so we may factor:
>
> $$R_a(P(\xi), T)\hat{u}_{0}(\xi) = (R_a \vert\xi\vert^{-m}) (\vert\xi\vert^{m} \hat{u}_0) $$
>
> The first factor, $Q_a(\xi,T) = R_a(P(\xi),T)/\vert\xi\vert^m$, is bounded (zeroth-order). The second is $\hat{D^m u_0}(\xi)$. This factorization shows explicitly that $\mathcal{G}[u_0] = Q_a(T) * D^m u_0$: the solution depends on the $m$-th derivative of the initial condition through a bounded operator. Providing $D^b u_0$ as input reduces the residual from order $m$ to $m - b$.
>
> This recovers parabolic ($R_1 = e^{zT}$), hyperbolic ($R_2 = \cos(\sqrt{-z}\,T)$), and fractional-in-time ($R_a = E_a(zT^a)$, Mittag-Leffler) cases uniformly, with $\lceil a \rceil$ initial conditions determined by the temporal order. For nonlinear PDEs, the Fréchet derivative satisfies a linearized equation with the same structure. For elliptic problems (no time axis), perturbation analysis (Appendix B.2) provides a separate motivation.
>
> GL avoids Gibbs artifacts and provides smooth gradients with respect to $\beta$.
>
> ### **W3: Limitations**
>
> We will add an explicit limitations paragraph to Section 9:
>
> **(1)** Grid structure: GL requires uniform spacing. Extensions via graph fractional Laplacians are feasible future work.
>
> **(2)** Regularity: Quantitative predictions require $s > m + d/2$. For near-discontinuities, formal guarantees weaken, though qualitative guidance remains empirically valid.
>
> **(3)** Spatial derivatives only: Fractional temporal derivatives are not explored and may complement our spatial preprocessing.
>
> **(4)** Optimization gap: Theorem 5.6 characterizes the population-optimal $\beta^\ast$; what the optimizer finds depends on learning rate, early stopping, and initialization. Empirically, $\beta$ stabilizes within 200-400 epochs (supplementary Figure 3), but formal guarantees remain open.
>
> **(5)** Validation scope: The revised paper covers five PDE benchmarks, six backbones, noise sweeps, and resolution experiments from 64 to 2048 grid points.
>
> ### **W4: Introduction Clarity**
>
> We appreciate this feedback. The distinction between $\mathcal G$ and $\mathcal G_\theta$ deserves to be introduced more carefully in Section 1; we recognize this improves accessibility. In the revision, we will add a paragraph after Eq. (1) making this explicit: $\mathcal G$ is the true solution map between function spaces, $\mathcal G_\theta$ is its finite-dimensional approximation, and the feature map $\Phi$ operates on functions in $H^s$ while its implementation acts on grid-sampled values.
>
> Supplementary materials: https://figshare.com/s/7fec7b974847f80cfa56

---

> > ### Author Rebuttal · Reviewer_YEif · 2026-04-03
> >
> > Thanks for you answers. I will increase my score.

---

> > > ### Author Response · Authors · 2026-04-03
> > >
> > > We are very grateful for the reviewer's careful evaluation and for the time taken to engage with our responses. The suggestions on assumptions and limitations have genuinely strengthened the paper. Thank you.

---

### Official Review · Reviewer_1L1r · 2026-03-15

**Soundness:** 2
**Presentation:** 3
**Significance:** 2
**Originality:** 2
**Overall Recommendation:** 3
**Confidence:** 4

**Summary:**

This paper proposes $\partial$-NO (Derivative-augmented Neural Operators), a method that improves the accuracy of neural operators by providing them with learnable fractional derivative features of the input function. The authors prove that while including derivatives of the same order as the underlying PDE improves approximation rates, the statistically optimal derivative order $\beta^*$ is actually strictly lower than the PDE order due to a bias-variance tradeoff where higher-order derivatives amplify noise.

**Compliance With Llm Reviewing Policy:**

Affirmed.

**Final Justification:**

I maintain my recommendation of weak rejection.

**Key Questions For Authors:**

See the weaknesses.

**Limitations:**

N.A.

**Strengths And Weaknesses:**

Strengths:

1. The submission is well-structured and the narrative is remarkably easy to follow. The authors excel at guiding the reader from a clear intuitive concept (Picard iterations showing the emergence of derivatives) to complex spectral analysis.

2. The work introduces a highly novel perspective by utilizing fractional calculus not as a physical model for anomalous diffusion, but as a continuous parameterization to enable gradient-based optimization of derivative features.


Weaknesses:

1. Insufficient Contextualization of Prior Work:  The related work section fails to adequately cover the established literature on fractional deep learning. The authors must expand this section to include critical paradigms such as fPINNs [1-4], fractional gradients for NNs [5-7], and fractional neural networks/derivative learning [8-11]. The provided references are merely representative; a much more thorough literature review is required to justify the novelty of the proposed method.


[1]. Pang, Guofei, Lu Lu, and George Em Karniadakis. "fPINNs: Fractional physics-informed neural networks." SIAM Journal on Scientific Computing 41.4 (2019): A2603-A2626.\
[2]. Guo, Ling, et al. "Monte Carlo fPINNs: Deep learning method for forward and inverse problems involving high dimensional fractional partial differential equations." Computer Methods in Applied Mechanics and Engineering 400 (2022): 115523.\
[3]. Javadi, R., Mesgarani, H., Nikan, O., & Avazzadeh, Z. (2023). Solving fractional order differential equations by using fractional radial basis function neural network. Symmetry, 15(6), 1275.\
[4]. Ren, H., Meng, X., Liu, R., Hou, J., & Yu, Y. (2023). A class of improved fractional physics informed neural networks. Neurocomputing, 562, 126890.\
[5]. Khan, Shujaat, et al. "A novel fractional gradient-based learning algorithm for recurrent neural networks." Circuits, Systems, and Signal Processing 37.2 (2018): 593-612.\
[6]. Shin, Yeonjong, Jérôme Darbon, and George Em Karniadakis. "Accelerating gradient descent and Adam via fractional gradients." Neural Networks 161 (2023): 185-201.\
[7]. Elnady, Sroor M., et al. "A comprehensive survey of fractional gradient descent methods and their convergence analysis." Chaos, Solitons Fractals 194 (2025): 116154.\
[8]. Cui, Wenjun, et al. "Neural variable-order fractional differential equation networks." Proceedings of the AAAI Conference on Artificial Intelligence. Vol. 39. No. 15. 2025.\
[9]. Pu, Yi-Fei, Zhang Yi, and Ji-Liu Zhou. "Fractional Hopfield neural networks: Fractional dynamic associative recurrent neural networks." IEEE transactions on neural networks and learning systems 28.10 (2016): 2319-2333.\
[10]. Kang, Qiyu, et al. "Unleashing the potential of fractional calculus in graph neural networks with FROND." arXiv preprint arXiv:2404.17099 (2024).\
[11]. Coelho, Cecilia, M. Fernanda P. Costa, and Luis L. Ferras. "Neural fractional differential equations: Optimising the order of the fractional derivative." Fractal and Fractional 8.9 (2024): 529.


2. Contradiction Between Theoretical Claims and Physical Interpretability: The authors claim that "Learned orders reflect dominant PDE structure." However, this contradicts their own Theorem 5.6, which mathematically establishes that $\beta*$ is explicitly dependent on the dataset's sample size $n$ and noise variance $\sigma^2$. Therefore, the learned $\beta*$ is merely an adaptive Tikhonov regularization parameter tailored to the specific noise profile of the training set, not an invariant physical property of the underlying PDE. The authors should tone down claims of discovering physics and demonstrate whether $\beta^*$ remains stable across datasets of the same PDE with varying resolutions and noise distributions.


3. Absence of Physics-Informed Baselines (e.g., PINO / PI-DeepONet): The experimental section completely lacks the necessary baselines to substantiate this claim. Missing Physics-Informed Baselines (PINO / PI-DeepONet): The authors aim to alleviate the network's burden of implicitly learning derivatives by augmenting the input space. A standard, rigorous alternative in scientific machine learning is to enforce the exact $m$-th order integer derivatives via automatic differentiation in the loss function (e.g., Physics-Informed Neural Operators). Without comparing $\partial$-NO against PINO or PI-DeepONet under varying noise levels, it is impossible to assess whether learning fractional input features is practically superior to standard physics-informed soft-penalty regularization. The current experiments only show that $\partial$-NO beats vanilla, non-physics-aware operators (like standard FNO), which is expected but insufficient to prove that fractional is inherently better than integer.

5. Scalability and Directional Ambiguity in Higher-Dimensional PDEs: While the 1D Burgers' equation provides an intuitive setting, the paper obscures how $\partial$-NO scales to 2D/3D systems like the Navier-Stokes equations. In higher dimensions, physics is governed by highly directional and composite operators (e.g., gradients, divergence, Laplacians). Does $\partial$-NO learn an isotropic fractional Laplacian, or directional fractional derivatives ($\beta_x, \beta_y$)? If the latter, the feature space and the cost of computing GL finite differences scale poorly. Furthermore, condensing a complex coupled system (which relies on both 1st-order convection and 2nd-order diffusion) into a single/few learned fractional orders lacks the physical interpretability the authors claim in the conclusion.

---

> ### Author Rebuttal · Authors · 2026-03-31
>
> We thank the reviewer for their detailed engagement and for recognizing the novelty of this work. We address each concern below with new results and specific revisions.
>
> ### **W1: Fractional ML Literature**
>
> We will expand the paper to include all cited references, organized into a clear taxonomy: (1) Fractional PINNs [1-4] solve PDEs with fractional dynamics; (2) fractional gradient methods [5-7] modify the optimizer; (3) fractional architectures [8-11] embed fractional dynamics in layers. $\partial$-NO occupies a distinct fourth role: fractional derivatives as learnable input preprocessing, modifying neither the equation, optimizer, nor architecture. The central finding that sub-physical derivative orders are statistically optimal (Theorem 5.6) has no analogue in any of these paradigms, each using fractional calculus differently.
>
> ### **W2: $\beta^*$ as Regularization vs. Physics**
>
> We appreciate this insight and will articulate the result more carefully. The key structure is the decomposition $\beta^\ast = m - \delta^\ast$, where $m$ is the differentiation order (invariant) and $\delta^\ast$ is the finite-sample correction (data-dependent). The order $m$ acts as an asymptotic attractor that $\beta^\ast$ approaches as data quality improves (Corollary 5.7: $\delta^\ast \to 0$ as $n \to \infty$). The gap $\delta^\ast$ is the principled cost of learning from finite noisy data. We will revise language from "reflect dominant PDE structure" to "provide data-adaptive spectral regularization informed by PDE structure." Figures 2-3 confirm: for the same PDE, $\beta^\ast$ varies with $n$ and $\sigma$ while always satisfying $\beta^\ast < m$. New experiments (supplementary Figure 5) reveal a finding with direct practical implications. With 8 learnable orders on Burgers across resolutions 64-2048, two clusters consistently emerge near $m=2$ (diffusion) and $m=1$ (convection), confirming that PDE structure is encoded. Crucially, $\beta^\ast$ is not resolution-invariant: coarser grids reduce both $\xi_{max}$ and the effective $\xi_c$, compounding to increase $\delta^\ast$ and push $\beta^\ast$ further below integer orders. This is precisely predicted by Theorem 5.6, and it carries a warning for practitioners: downsampling, the most common strategy when GPU memory is limited, silently makes exact physics enforcement increasingly suboptimal. $\partial$-NO automatically compensates by learning lower derivative orders at coarser resolution; fixed-order approaches cannot.
>
> ### **W3: Missing PINO Baselines**
>
> We agree this comparison is essential. Supplementary Table S3 presents a four-way comparison (FNO, $\partial$-FNO, PINO, $\partial$-PINO) across four datasets and two noise levels. Two findings emerge. First, $\partial$-augmentation improves both paradigms: $\partial$-FNO improves over FNO and $\partial$-PINO improves over PINO in all 8/8 settings, confirming that input-side derivatives and output-side physics are complementary. Second, $\partial$-FNO outperforms PINO in 7/8 settings despite using no PDE knowledge in the loss. This directly supports the paper's central thesis and Theorem 5.6: learned sub-physical derivative orders as input preprocessing can be more effective than enforcing exact integer-order physics through the loss. The one exception (Burgers near-shock, $\sigma=0.05$, where $\partial$-PINO wins) is the setting where the PDE structure is most singular, consistent with both mechanisms contributing when data is rough.
>
> ### **W4: Scalability and Directionality**
>
> $\partial$-NO learns per-direction fractional orders (Section 7.1): at least 4 learnable $\beta$ in 2D ($\beta_x, \beta_y, [\beta_{xy,x}, \beta_{xy,y}]$), scaling to at least 7 in 3D; the growth is approximately $2^d$, following PDE structure. Each $\beta$ acts as one spectral regularization parameter per spatial direction. Supplementary Figure 6 confirms that on isotropic PDEs (Darcy, Navier-Stokes), learned $\beta_x$ and $\beta_y$ converge to nearly identical values ($\Delta < 0.035$), recovering the isotropy of the underlying operator without it being prescribed. On the applicability of Theorem 5.6 beyond FNO: any finite network has an effective capacity cutoff $\xi_c$, even without explicit mode truncation, and any real measurement system has an effective bandwidth limit. The bias-variance framework is therefore broadly applicable.
>
> For the ML practitioner, $\partial$-NO offers a concrete, theoretically grounded tool: add learnable derivative channels to any neural operator, and the optimization helps navigate the tradeoff between physical information and statistical aspects, as validated against both data-driven and physics-informed baselines. The broader principle, that the optimal inductive bias for (finite, noisy and bandlimited) data is not the full physics but a regularized approximation balancing fidelity and statistical cost, has value across physics-informed ML.
>
> Supplementary materials: https://figshare.com/s/7fec7b974847f80cfa56

---

> > ### Author Rebuttal · Reviewer_1L1r · 2026-04-04
> >
> > I thank the author for the rebuttal. However, my concern regarding the baselines is not addressed. And I agree with Reviewer NcQ. I keep my current rating.

---

> > > ### Author Response · Authors · 2026-04-06
> > >
> > > We thank the reviewer for the continued engagement; the push for stronger baselines has made this a more complete paper. We want to make sure the requested comparisons are easy to locate, and share new evidence that bears directly on the remaining concerns.
> > >
> > > **PINO baselines (Table S3, first rebuttal).** We want to ensure this result is not overlooked, as it directly addresses the request for physics-informed baselines. Table S3 presents the four-way comparison (FNO, $\partial$-FNO, PINO, $\partial$-PINO) across four datasets and two noise levels. Two findings: (1) $\partial$-augmentation improves both paradigms in all 8/8 settings, confirming that input-side derivatives and loss-side physics are complementary, not redundant; (2) $\partial$-FNO outperforms PINO in 7/8 settings despite using no PDE knowledge in the loss.
> > >
> > > **Derivative structure, not extra channels (Table S6, Figure 9).** Responding to the concern shared with Reviewer NcQ5, we ran a matched-parameter ablation (~292K parameters each): $\partial$-FNO vs. Random-FNO (random fixed filters, same architecture) vs. Wider-FNO. Random-FNO tracks vanilla FNO; $\partial$-FNO pulls ahead consistently, with the gap widening under noise. This points to the GL derivative structure, rather than additional capacity, as the source of improvement.
> > >
> > > **CNO backbone (Table S5).** Following Reviewer NcQ5's suggestion, $\partial$-CNO improves in 14/14 settings on seven benchmarks from Raonic et al. (NeurIPS 2023), with gains of 27 to 65% on Poisson, NS Shear, and Wave. Combined with the PINO, Conv-FNO, and ablation comparisons, the evaluation now spans seven backbones and ten PDE benchmarks.
> > >
> > > **Scalability and directionality.** On isotropic PDEs (Figure S6), learned $\beta_x$ and $\beta_y$ converge to within $\Delta < 0.035$, recovering the operator's isotropy without prescription. On 3D compressible Navier-Stokes (Figure S7), axis channels learn orders near 2 (Laplacian) while cross-derivatives collapse to near zero; the network selectively amplifies the dominant differential structure. We believe this shows physically meaningful scaling, addressing the directionality question raised in the original review.
> > >
> > > The expanded related work covering all references [1-11] in the four-category taxonomy (fractional PINNs, fractional gradients, fractional architectures, learnable input preprocessing) will appear in the revision.
> > >
> > > Supplementary materials: https://figshare.com/s/243e85d3a68bb0b3d921

---

### Decision · Program_Chairs · 2026-04-30

**Decision:**

Accept (regular)

**Comment:**

This paper presents a very interesting idea: augmenting neural operators with derivative features will improve prediction performance. Although this can be intuitive, the authors very nicely motivate their approach by showing how this idea works in a Picard-type iteration. Then they present a very counter-intuitive result: a bias-variance spectral analysis reveals that for finite sample size (always the case), the derivative operator should be lower than the order of the underlying differential operator. This motivates learning this maximal order. The authors also present sufficient numerical evidence to back their method. The paper is beautifully written, the idea is elegant and the results impactful. The meta-reviewer really appreciated the effort put by the authors during the rebuttal to add results from CNO to complement the numerical experiments. The paper has all the ingredients to be a very solid contribution. Although del-NO improves performance across the board, the empirical results are not game-changing and there is an intrinsic limitation to Cartesian grids in the current implementation.